behaviour/computer vision/optics

mosquito tracking, retro-reflective screen, entomology

**Author for correspondence:**
Vitaly Voloshin
e-mail: v.voloshin@warwick.ac.uk

# Diffuse retro-reflective imaging for improved video tracking of mosquitoes at human baited bednets

Vitaly Voloshin[1], Christian Kröner[1],
Chandrabhan Seniya[1], Gregory P. D. Murray[2], Amy Guy[2],
Catherine E. Towers[1], Philip J. McCall[2] and
David P. Towers[1]

[1]School of Engineering, University of Warwick, Coventry, CV4 7AL, UK
[2]Liverpool School of Tropical Medicine, Pembroke Place, Liverpool, L3 5QA, UK

VV, 0000-0002-9631-5976; CET, 0000-0001-8296-5540;
DPT, 0000-0002-2618-1626

Robust imaging techniques for tracking insects have been essential tools in numerous laboratory and field studies on pests, beneficial insects and model systems. Recent innovations in optical imaging systems and associated signal processing have enabled detailed characterization of nocturnal mosquito behaviour around bednets and improvements in bednet design, a global essential for protecting populations against malaria. Nonetheless, there remain challenges around ease of use for large-scale *in situ* recordings and extracting data reliably in the critical areas of the bednet where the optical signal is attenuated. Here, we introduce a retro-reflective screen at the back of the measurement volume, which can simultaneously provide diffuse illumination, and remove optical alignment issues while requiring only one-sided access to the measurement space. The illumination becomes significantly more uniform, although noise removal algorithms are needed to reduce the effects of shot noise, particularly across low-intensity bednet regions. By systematically introducing mosquitoes in front of and behind the bednet in laboratory experiments, we are able to demonstrate robust tracking in these challenging areas. Overall, the retro-reflective imaging set-up delivers mosquito segmentation rates in excess of 90% compared to less than 70% with backlit systems.

# 1. Introduction

Many arthropod vectors of human infections are highly adapted to the human home, and, in many regions worldwide, transmission of malaria, leishmaniasis, Chagas disease, lymphatic filariasis and tick-borne relapsing fever occurs when the vectors take blood from sleeping humans. Bednets treated with insecticide can be very effective in preventing transmission of these infections, and, in Africa, the factory-treated durable type of nets referred to as long-lasting insecticidal nets (LLINs), are the most effective method available and an essential element of malaria control and elimination strategies today. Increased understanding of vector behaviour at LLINs, including how different LLIN treatments affect mosquito flight around, and contact with, the net surface is essential for understanding LLIN modes of action [1,2] and raises the prospect of developing more effective interventions to reduce disease transmission [3,4].

Optical imaging techniques have been used for decades in entomological studies in diverse settings in the laboratory and field, and recently a tracking system was used to characterize in detail how *Anopheles gambiae* mosquitoes, the vectors of human malaria in sub-Saharan Africa, interact with human-occupied bednet [1,2,5]. Offering high spatial and temporal resolution, optical imaging has clear advantages for quantifying vector behaviour [6,7], but its application in typical sub-Saharan Africa dwellings present particular challenges. In Africa, insecticidal bednets are by far the most effective intervention available for malaria control and the most widespread method used to prevent its transmission [8–10]. Sustaining the required high level of efficacy against increasingly insecticide-resistant vector populations requires novel insecticide treatments on bednets, and vector biologists need to understand how hungry host-seeking mosquitoes interact with different treatments, or how net alterations (physical as well as chemical) might alter efficacy. To investigate this, they are particularly interested in visualising activity around the sleeping human body, the bednet suspended above it and the regions around this in order to examine details of approach, attack and departure [11,12]. Hence, the inspected volume ideally needs to be $2 \times 2.5 \times 1.5$ m (depth × width × height), which generously encompasses the space around a typical installed bednet. To avoid rapidly varying spatial resolution between the front and back of the measurement volume, telecentric approaches are desirable for both illumination and imaging, and, given the space constraints in sub-Saharan dwellings, this leads to typically 0.5 numerical aperture (NA) optical systems. The bednet itself is a regular grid of polyester or polyethylene fibre (typically 75–180 denier, mesh size of 24–32 holes $cm^{-2}$) that can easily occlude the images of a slender unfed *Anopheles gambiae* female (body length and thickness typically less than 10 and 3 mm, respectively) [13,14]. Hence the visibility (contrast) of the mosquito's image is reduced when the mosquito is in front of or behind a bednet, as the bright background field is attenuated from transmission through the bednet layers each side of the human bait. Furthermore, for field studies the optical system needs to be simple to transport and install, and sufficiently robust to withstand environmental instabilities, e.g. flexible wooden floors, as well as computationally efficient to extract the flight tracks required by entomologists.

Two- and three-dimensional (multi-camera) imaging set-ups have been reported using illuminated diffuse surfaces or lamps as a background [15,16]. Combined with algorithms for tracking, two- or three-dimensional flight trajectories are produced, from which responses to attractants or interventions can be determined via manual inspection. However, such analyses, while of value, cannot determine responses to an insecticide-treated bednet because of the inability to capture large enough volume. Field studies to examine mating behaviour have been reported using the setting sun as a back light with a pair of stereo cameras to give three-dimensional mosquito tracks over volumes of metre-scale dimensions, but this imaging approach cannot be translated to the inside of dwellings in nocturnal situations [17–19]. Stereo or multi-camera three-dimensional imaging provides spatial resolution that increases proportionately with the field of view [20,21]. Another multi-camera approach (up to 11 cameras) for three-dimensional tracking flying animals was reported in [22]. The tracking in [22] relies on high levels of spatial sampling in order to extract position and orientation information of the animal as well as on the use of a network of processing computers (up to nine computers).

Millimetre-scale resolution should be available over room-size volumes from optically suitable surfaces, but performance will degrade with mosquito targets that vary in presentation according to angle of view of the multiple cameras. Furthermore, a minimum of two camera views are needed in each region for three-dimensional metrology, hence to adequately map the space around a human baited bednet would require pairs of cameras for each side, the head and feet areas as well as cameras to map the space above the bednet. The entirety of the bednet surface needs to be captured as this is where mosquitoes interact with the insecticide. Hence, the test room would need to have five sides largely transparent in order to position the cameras outside of the room and look in (meaning modifications to the roof region where mosquitoes are known to enter via eaves), or a significantly larger sized building that would be atypical compared to sub-Saharan dwellings. Moreover, multi-camera three-dimensional systems require significant levels of processing power that is often not available in the field.

Large field of view backlit imaging systems which record the whole bed with bednet and surrounding areas have been reported [5]. The set-up uses two parallel imaging channels (two non-intersecting camera images) to give a measurement volume of $2 \times 2 \times 1.4$ m in total. Large aperture Fresnel lenses enable collimated illumination and telecentric imaging. Illumination was provided by a single LED with a transmission diffuser located behind the back Fresnel lens for each camera. As part of this study, algorithms were also reported that produced flight tracks of 25 mosquitoes over recording periods of up to 2 h. The recording system enabled discrimination of four behavioural modes during the mosquito's interactions with a human-occupied bednet: non-contact flights (swooping), and flights with single (visiting), multiple rapid (bouncing) or sustained (resting) net contact [5]. This became the basis for elucidation of the mode of action of insecticidal bednets [8], studies that, in turn, led to the 'barrier bednet', an innovative and field-tested concept that could greatly expand the potential and lifespan of insecticidal bednets [4,23]. Despite these advances, conducting these experiments in the field is challenging. The imaging approach worked in transmission with a Fresnel lens at either end of the measurement volume, and consequently 1.5 m was needed beyond the lenses at each end. The use of two Fresnel lenses per camera also generates undesirable amplitude modulation in the images in circular rings and needs very careful alignment of the two lenses with respect to each other. The unstable nature of test environments in sub-Saharan Africa dwellings means that the alignment of these large aperture Fresnel lenses needed regular adjustment.

Here, we introduce an optical set-up using a retro-reflective screen (RRS) to eliminate the optical alignment problems, reduce the size of the optical system and improve the uniformity of the images. In this approach, a single Fresnel lens per camera is used to both collimate the illumination beam and focus the light reflected from the RRS at the far end of the measurement volume. No further optical components are needed beyond the RRS. Similarly to the backlit approach, the volume captured is $2 \times 2 \times 1.4$ m and two parallel imaging channels are used with one camera per channel. In transmission through a bednet, the light amplitude is reduced and in this reflective mode the light is attenuated through twice the number of bednet layers compared to the backlit set-up. Additional data processing steps are introduced to handle the increased range of contrast in the images, in particular, to manage the reduced contrast of the mosquito images partially obscured by layers of net. The following sections describe the optical set-up and signal processing. Sets of experimental data are shown where mosquitoes are introduced into known spatial locations—e.g. in front of or behind a bednet—as a means of confirming mosquito detection ability in all regions of the image. Hence, tracking performance comparisons are made between the backlit and RRS-based imaging approaches.

Finally, the RRS-based tracking system's potential for a range of behaviour studies is shown by its performance in a binary test of the relative attractiveness of two human hosts, where multiple mosquitoes are tracked as they orient to and select the individual on whom they will land and bloodfeed.

# 2. Methods

## 2.1. Optical set-up

The optical set-up for the RRS approach is shown in figure 1*b*, and, for comparison purposes, the backlit set-up is in figure 1*a*. Light from an LED ring light expands over the Fresnel lens (approx. $1.4 \times 1.0$ m aperture, 1.2 m focal length, NTKJ Co., Ltd, CF1200 [24]) and is then approximately collimated to illuminate the space above the bednet and through the bednet itself. A volunteer acts as human bait lying beneath the bednet. The telecentric set-up (created by the combination of camera and Fresnel lenses) plays a key role in image formation and metrology of mosquito position. As the depth of the scene is quite large, approximately 2 m, telecentric imaging is essential to determine the mosquito displacements accurately when located at different depths. Moreover, the collimated illumination and imaging enables neighbouring camera views to be independent and give sufficient spatial resolution over the measurement volume.

The light travels to a back wall where it is reflected back via a RRS. While a number of retro-reflective materials are available, improved performance was found using 3M™ Scotchlite™ High Gain 7610 [25] obtained as a tape to be stuck to the back wall. The material contains glass beads and is based on an exposed lens material, which means it does not have a glossy surface covering but a matte appearance producing a diffuse background with scattered light over a 10° reflection cone. The retro-reflective material applied to a plywood board (size $2.4 \times 1.2$ m) per camera is placed approximately 2 m from

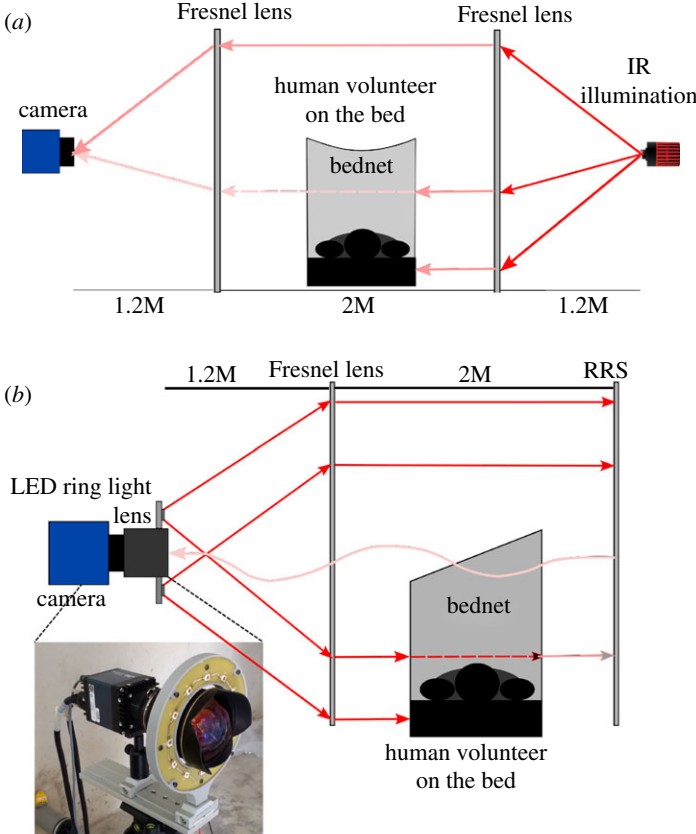

**Figure 1.** Schematic diagram of the original backlit and new retro-reflective screen recording systems. (*a*) Backlit recording set-up, (*b*) retro-reflective recording set-up with an inset photograph of the ring light mounted together with the camera on a standard photographic tripod.

the Fresnel lens. The reflected light re-crosses the measurement volume and is focused by the same Fresnel lens which forms a telecentric lens pair with an imaging optic mounted on the camera. This configuration allows illumination and imaging from one side of the bednet and scene and is relatively insensitive to alignment owing to the retro-reflective nature of the beads in the RRS.

Ideally, the illuminating LED would be placed on axis along with the camera optic; however, this is practically difficult owing to the high NA needed in both illumination and imaging at approximately 0.5. The LEDs are therefore positioned outside the camera optic's aperture leading to pairs of images from individual mosquitoes. A direct image of the mosquito is seen by the camera against the bright RRS and a shadow image is formed from the slightly off-axis LED on the RRS. This effect enables triangulation and hence recovery of three-dimensional mosquito position data [26] but requires complex calibration and extensive signal processing. By contrast, it has been demonstrated that two-dimensional tracking provides entomologically useful information [1,2,4], and the requirement in malaria control is to rapidly test design iterations of interventions over long time periods (hours) and with sufficient repeats to give statistically reliable data. Hence, this paper targets robust two-dimensional tracking.

For two-dimensional tracking over the 2 m depth of field needed here, only the direct image is required. The contrast of the direct image is higher than that of the shadow images due to the extended distance over which diffraction occurs for the latter. Furthermore, each LED within the ring light will form shadows in different spatial locations of the image, whereas the direct image is in the same location and becomes reinforced with each additional LED. A custom ring light source was constructed with 12 OSRAM™ SFH 4235 infrared LEDs (peak wavelength 850 nm) [27]. The wavelength provides good sensitivity for monochrome silicon-based detectors, and there is no evidence from previous backlit experiments or ongoing studies with the RRS set-up that the direction of the illumination and orientation of the human bait affected mosquito behaviour [1,6].

The inset in figure 1*b* shows a photograph of the ring light and mounting arrangement via an optical rail that allows independent adjustment of LED plane with respect to the camera and lens. RRS-based imaging experiments were conducted with 12 Mpixel cameras, either Ximea CB120RG-CM (used with

Canon EF 14 mm f/2.8L II USM lenses) or Dalsa FA-80-12M1H (used with Nikon 14 mm f/2.8D AF ED lens), operating at 50 frames s$^{-1}$ and with 14 mm focal length camera lenses. The optical system (camera lens, Fresnel lens and RRS screen) is approximately telecentric, with the deviations coming from the steps in the Fresnel lens (which makes such a large aperture lens physically manageable) and the RRS which gives a cone of reflected light, partly compensated by closing the camera lens aperture to typically F8.0. The optical set-up provides approximately 0.5 mm per pixel which gives adequate sampling of the *Anopheles gambiae* mosquito which is typically 2.5–3.5 mm wing length (other species can be much larger). Hence, in the images, there is a bright background with the mosquitoes appearing as dark images. In the RRS approach, there is a double pass of the measurement volume giving increased diffraction, which limits the practical depth of field to approximately 2 m.

## 2.2. Signal processing

In order to interpret mosquito behaviours it is imperative to be able to track them when swooping above or around the bednet as well as when host seeking and probing the net. To form contiguous tracks across all the different areas, a robust segmentation process is needed. When the mosquitoes are in the region above the bednet there is at least 3–5 greyscales difference between the mosquito image and the background. The contrast reduces markedly when the mosquito is either in front of or behind the bednet. With the RRS set-up, the light passes each layer of the bednet twice (figure 1*b*) leading to 2–3 greyscales difference between the mosquito and background when the mosquito is in front of the net (closer to the Fresnel lens) and about 1–2 greyscales for mosquitoes behind the net. Graphs of the typical intensity distribution around a mosquito image are given in figure 2 in both three-dimensional (inverted intensity scale) and image formats. Figure 2*a* shows a typical mosquito shadow image using backlit imaging when the mosquito is outside the bednet, the corresponding image with RRS imaging is in figure 2*b*. Significantly reduced contrast can be seen in figure 2*c* which is a backlit mosquito image from across the bednet and figure 2*d* from the RRS approach. Mosquito detection relies on the contrast between the bright background and dark image of the foreground (mosquitoes). Objects that do not allow light transmission (e.g. the human host, the bed, etc.) prevent mosquito detection in front of or behind such objects similarly for both backlit and RRS set-ups.

The original approach to segmentation [5] used a difference image between consecutive frames and a single threshold value to identify movement. While developing the RRS system, two issues were found. Firstly, the contrast of a mosquito image in low-intensity areas is similar to the camera noise. Secondly, in high-intensity regions, the noise level and contrast is higher, necessitating a different threshold. Noise reduction is necessary to enable robust thresholding of mosquito images in low-intensity areas. Several filter types were considered (Gaussian, median, etc.) and applied to original or difference images. Using noise reduction effectiveness and computational overhead as criteria, a Gaussian filter applied to the difference image was selected using a $15 \times 15$ pixel kernel (implemented via the OpenCV library [28]). The width of the Gaussian function is defined by the standard deviation as [28]

$$\sigma = 0.3 \cdot ((ksize - 1) \cdot 0.5 - 1) + 0.8.$$

The filter values were scaled such that the integral across the whole filter was 1. The typical benefit of the filter can be observed in figure 3 which shows as red the pixels above a user-defined threshold before and after application of the filter, in this example, the threshold was 2 greyscales.

To address the need for variations in threshold to segment a mosquito image, the high- and low-intensity areas can be classified from examination of a normalized histogram. For the RRS system, the greyscale ranges associated with the different regions of the image can be seen in figure 4*b*. The two regions of interest for mosquito flight are the background, which is above or to the side of the bednet and the bednet region itself, either in front of or behind the bednet. A manual definition of the threshold that separates these regions is required to run the segmentation algorithm and is typically around 40–60 greyscales and is consistent for both backlit and RRS-imaging approaches.

The system is insensitive to the ambient lights level (if it does not create direct reflection on the Fresnel lens). Thresholding is applied to the difference image obtained between nearby frames in the temporal sequence, hence the threshold level is largely insensitive to variations in ambient light. The image obtained is dominated by the IR illumination directed to the RRS and hence reflected into the camera. Furthermore, field experiments are usually done at night, with closed windows and doors, and hence the threshold required for successful segmentation is stable. So, ambient light level encountered in typical experiments cannot affect the selected thresholds.

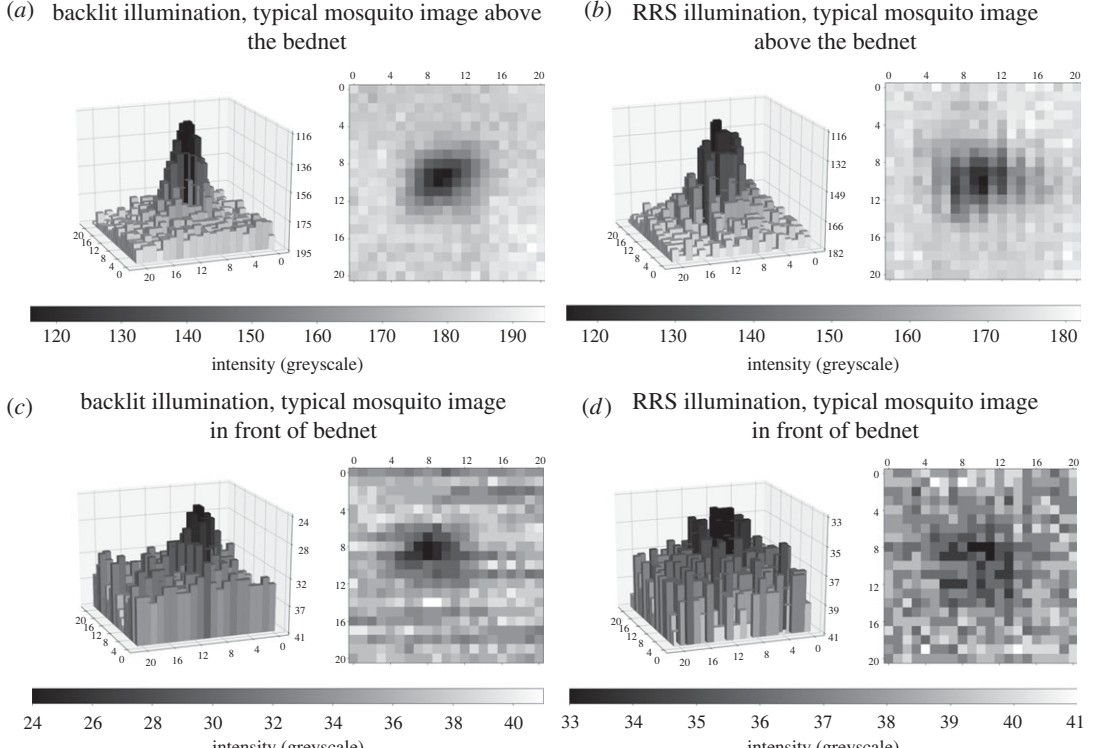

(a) backlit illumination, typical mosquito image above the bednet

(b) RRS illumination, typical mosquito image above the bednet

intensity (greyscale)

intensity (greyscale)

(c) backlit illumination, typical mosquito image in front of bednet

(d) RRS illumination, typical mosquito image in front of bednet

intensity (greyscale)

intensity (greyscale)

**Figure 2.** Mosquito intensity distributions under backlit or RRS illumination and for mosquito positions above the bednet or in front of the bednet. Three-dimensional maps use inverted intensity scale. (a) Backlit illumination, typical mosquito image above the bednet, (b) RRS illumination, typical mosquito image above the bednet, (c) backlit illumination, typical mosquito image in front of bednet, (d) RRS illumination, typical mosquito image in front of bednet.

(a) image before denoising applied

(b) image after de-noising, 15 x 15 pixel Gaussian filter, standard deviation 2.6

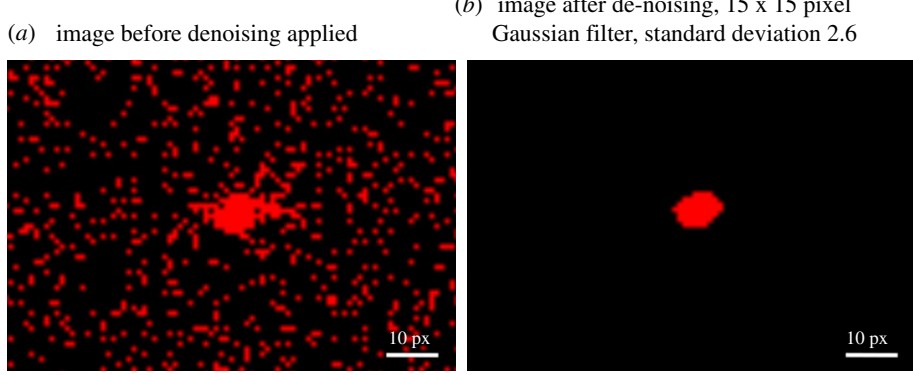

10 px

10 px

**Figure 3.** The typical effect of denoising on part of a difference image containing a mosquito image. Both cases show the same part of the frame. The red colour shows differences greater than 2 greyscales. (a) Image before denoising applied. (b) Image after de-noising, $15 \times 15$ pixel Gaussian filter, standard deviation 2.6.

It can be seen from figure 4 that the RRS system gives larger regions of high intensity in the background whereas this region is more widely distributed for the backlit case. The bednet region has a larger greyscale range for the backlit system (showing less uniform illumination in the backlit case), but its peak is at a higher greyscale, approximately 40, than for the RRS system. It can also be seen that the darkest regions, corresponding to the human bait are brighter for the RRS by *ca* 10 greyscales than in the backlit case. This occurs due to the direct front illumination which will also be incident on the mosquitoes and can slightly reduce the contrast in comparison with the background.

The updated segmentation algorithm incorporates the following stages.

(i) Form a difference image between the $n$th and $(n − i)$th images. Typically, $i$ is set to five frames to aid discrimination. This produces a pair of detected mosquito images with opposite signs and their separation is the mosquito displacement between the two frames.

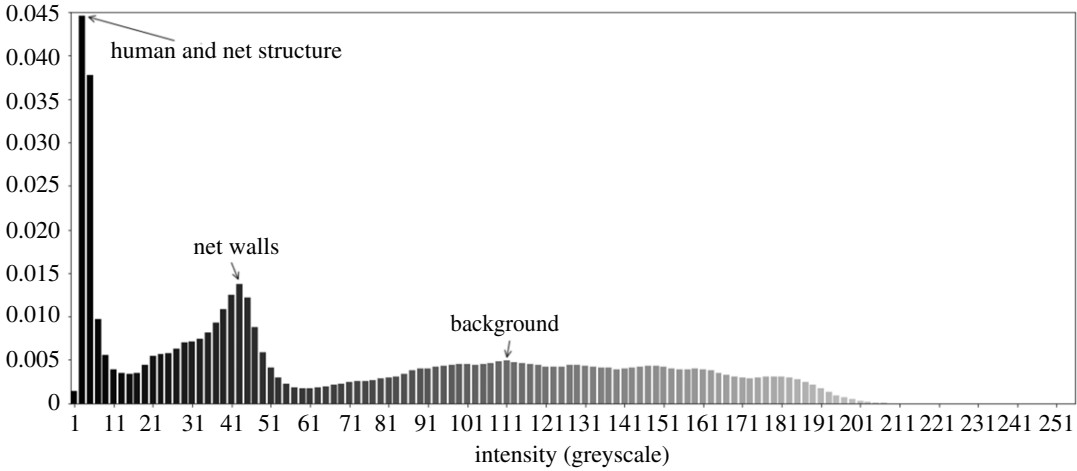

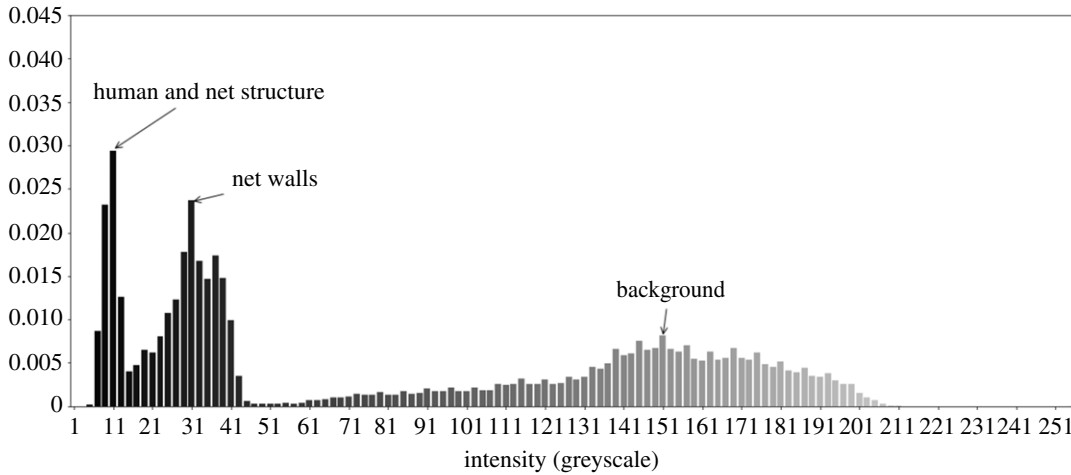

**Figure 4.** Normalized image histograms with different illumination approaches. (*a*) Backlit image histogram, (*b*) RRS image histogram.

(ii) A large kernel, typically $9 \times 9$ to $21 \times 21$ pixel, Gaussian filter is then (optionally) applied to the difference image to reduce noise.

(iii) For the brighter, background region, the algorithm searches for all regions of positive difference. An image threshold is calculated and applied to segment the mosquito image. Morphological operators can be optionally applied to improve the outer contour of the mosquito image (opening and closing). Statistics are then calculated: the centre coordinates, the area, i.e. standard 'blob' analysis. Dilation performed by 5–7 pixels aid the detection of broken-down images. Mosquito images are removed with either too large or small area (based on user-defined thresholds). The result is a set of candidate mosquito images that satisfy the selection criteria and their positions in the image. The typical threshold used for the bright areas is 3 greyscales.

(iv) The process is repeated for the darker image regions containing the bednet. The threshold used to segment the mosquito images tends to be lower at 1 to 2 greyscales.

(v) The data from the two passes (the brighter and darker image regions) are then combined together.

Apart from the addition of a Gaussian filter for de-noising and classification of the region to determine the appropriate threshold, the mosquito segmentation algorithm is identical to the one described in [5]. While more complex algorithms could be implemented, the requirement to control the computational overhead was important given the need for operation in the field and on datasets of approximately 2 TB per hour of recording from the two cameras.

# 3. Results and discussion

## 3.1. Evaluation of Illumination quality

Exemplar images from the backlit and RRS illumination systems are given in figure 5. The ring structure from the pair of Fresnel lenses used in the backlit approach is visible across the image in figure 5a, whereas this structure is averaged out when a single Fresnel lens is used with the RRS in figure 5b. The horizontal banding in the RRS image derives from the 50 mm wide adhesive tape obtained from 3M and the reflections from the LED ring light give small localized areas where individual mosquito images cannot be segmented. In the backlit case, the roof of the bednet is tilted whereas with RRS the data were obtained using a flat bednet roof. Each figure shows three intensity profile graphs in the region above the bednet, i.e. the background. It can be observed that with the RRS the intensity towards the edge and corners of the image is maintained at a higher level, whereas with backlit imaging the intensity drops to less than 50% of the central area. It is clear that in the corners of the frame from the backlit system the illumination level is very similar to that in the middle of the bednet (figure 6a). This will cause reduced mosquito detection in the corners of the frame and difficulty in defining a robust threshold.

The ring structure from the two Fresnel lenses can also be seen in the profiles from figure 5a with backlit imaging and for unstable field recordings can lead to erroneous artefacts in the difference images used for mosquito segmentation.

The temporal stability of the illumination using the bespoke LED ring light source was compared with that of the original single LED used in the original backlit system. Statistical analysis over 250 frames (5 s) of data in different parts of the image showed the expected result that the standard deviation and greyscale range increase with mean intensity. The cameras used for backlit imaging and those in the RRS experiments both using the same type of CMOS detectors and gave average levels of 1.84% greyscale standard deviation as a proportion of the mean intensity.

## 3.2. Quantitative performance metrics

The performance of the combined imaging and segmentation algorithms has been quantified using a number of metrics, thus enabling a quantitative comparison of the backlit and RRS imaging set-ups. The data are presented in table 1. Each dataset was obtained from a typical 1 hour recording with the cameras operating at 50 frames s$^{-1}$.

The recording made with the backlit system was done using Tiassale strain (an insecticide-resistant variety) of *Anopheles gambiae* mosquitoes. There were 25 female mosquitoes, between 3 and 5 days old, unfed and deprived of sugar for 4 h prior to filming and left to acclimatize in the filming room for 1 h before testing. The test was conducted within 1–3 h of the start of scotophase (i.e. their night time). The male human host was clothed, barefoot and lay on his back and as immobile as comfortably possible. This test was done at 28°C and 78% relative humidity (RH).

The recording with RRS system was done using N'guosso strain (an insecticide-susceptible variety) of *Anopheles coluzzii* mosquitoes. There were also 25 mosquitoes, between 3 and 7 days old, deprived of sugar 1 day before and of water 5 h before testing. Mosquitoes were placed in the room 1 h before testing to acclimatize. Mosquitoes were kept and the test was done in the same climate, i.e. at 27 ± 2°C and 70 ± 10% RH under a 12 h light/12 h dark cycle.

Both tests used modified PermaNet 2.0 LLIN bednets (55 mg m$^{-2}$ deltamethrin; Vestergaard, Lausanne, Switzerland). The modification involved making the roof tilted, so that the roof will be visible in the view of the camera. In both cases, all mosquitoes were released at the beginning of the test from a cup located 2 m above the floor and about 1.5 m from the edge of the bednet.

The same recording (made with RRS recording system) was used for the figure 7.

It is important to emphasize that the same algorithms have been used to process the recordings from both the backlit and RRS systems (with small variations in parameters to optimize the results). Therefore, differences in performance are due to the optical set-up. Notably, the greyscale threshold for detection of mosquito images across the challenging bednet region was the same (1 greyscale). An initial tracking process was performed [5] allowing only a single missing mosquito position within a track and discarding any tracks with less than five positions; hence the number of 'detected positions in tracks' in table 1 is determined. A second tracking iteration enabled existing tracks to be lengthened and tracks to be combined together with gaps of up to 15 frames between consecutive positions in a track.

(*a*)  image from backlit system  (*b*)  image from retro-reflective screen system

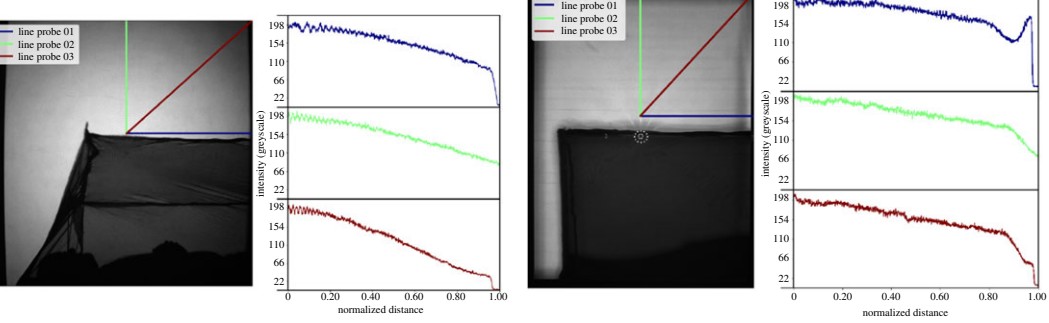

**Figure 5.** Spatial distribution of light levels across the background of the image along three different directions using line probes. Line probes are parametrized by a coordinate from 0 to 1, where 0 is at the centre of the image and 1 is at the edge. (*a*) Image from backlit system, (*b*) image from retro-reflective screen system.

(*a*)  image from backlit system  (*b*)  image from retro-reflective screen system

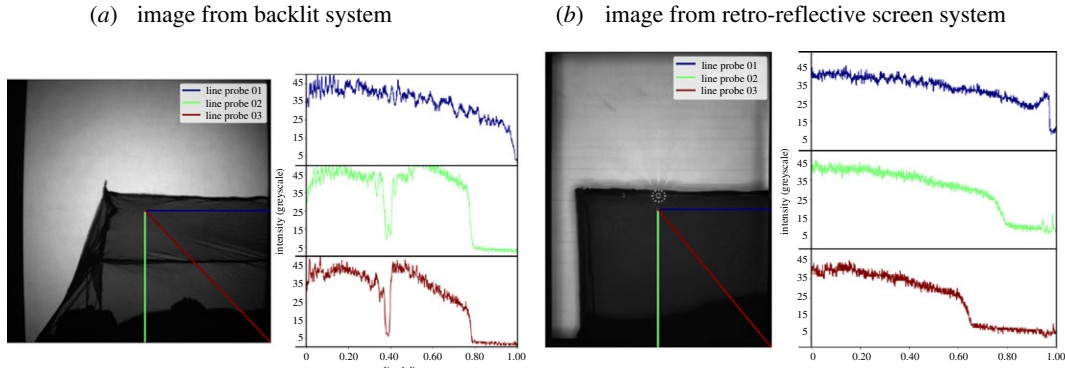

**Figure 6.** Spatial distribution of light levels across the bednet along three different directions using line probes. Line probes are parametrized by a coordinate from 0 to 1, where 0 is at the centre of the image and 1 is at the edge. (*a*) Image from backlit system, (*b*) image from retro-reflective screen system.

**Table 1.** Performance metrics of mosquito position detection for backlit and retro-reflective screen systems.

| systems | backlit [5] | RRS[a] |
|---|---|---|
| metrics | | |
| detected positions in tracks | 19 128 | 22 483 |
| mean $\pm$ s.d. gap size (frames) across net | 6.88 $\pm$ 11.91 | 4.19 $\pm$ 4.22 |
| gaps/positions ratio across net | 0.066 | 0.018 |
| mean $\pm$ s.d. gap size (frames) outside net | 4.18 $\pm$ 6.58 | 4.58 $\pm$ 4.62 |
| gaps/positions ratio outside net | 0.036 | 0.017 |
| average track length | 78.02 | 77.54 |
| average uninterrupted track length | 16.82 | 29.85 |
| detected mosquitoes across net | 54% | 92% |
| detected mosquitoes outside net | 85% | 92% |
| detected mosquitoes overall | 69% | 92% |

[a]Retro-reflective screen system.

Any gaps in a track, i.e. where the time between consecutive track positions is a multiple of the inter-frame time, $\Delta t$, indicate a missed mosquito position(s). Some missed positions occur as mosquitoes fly across completely occluded areas but others are due to locally poor contrast. The total number of

(*a*) positions and tracks of mosquitoes flying in front of the bednet

(*b*) positions and tracks of mosquitoes flying behind the bednet

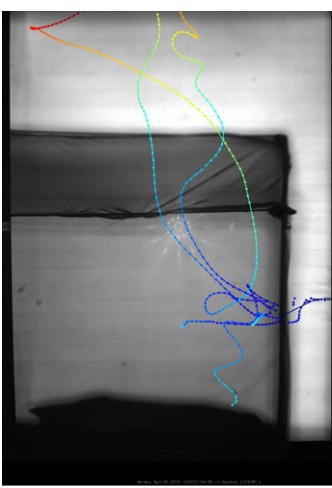
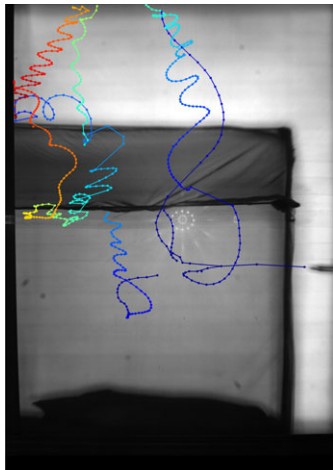

**Figure 7.** Segmented positions of the mosquitoes and flight tracks released using an aspirator (visible on the right of each frame) specifically in front of (*a*) and behind (*b*) the bednet. (*a*) Positions and tracks of mosquitoes flying in front of the bednet. (*b*) Positions and tracks of mosquitoes flying behind the bednet.

potential positions in a track, $N_t$, is calculated from

$$N_t = \sum_{k=0}^{n-1} \text{Int}\left[\frac{\Delta t_k}{\Delta t}\right],$$

where $n$ number of recorded positions, $\Delta t_k$ time difference between the current and previous recorded position, $\text{Int}[\dots]$ gives the nearest integer value. An individual gap in a track is identified when $\text{Int}[\frac{\Delta t_k}{\Delta t} - 1] > 0$ and hence the total length of gaps in a track, $N_{\text{Gaps}}$, is given by

$$N_{\text{Gaps}} = \sum_{k=0}^{n-1} \text{Int}\left[\frac{\Delta t_k}{\Delta t} - 1\right],$$

The data in table 1 provide average and standard deviation of gap size as well as the number of gaps as a proportion of the number of detected positions (to normalize the expected experimental variability). These metrics are determined for the background (outside the bednet) and the across the bednet regions. Statistics are also given for the average track length and average track length without gaps. The latter is the average length of track pieces without any gaps, i.e. where there is a detected position at every time step. Finally, mosquito detection percentages are given from the ratio between number of detected positions used in tracks and total number of potential positions, $N_t$.

## 3.3. Discussion

The rationale for the RRS-based imaging system has been described above, and it is clear that this approach delivers a smaller operational footprint as well as only needing optical access from one side of the measurement volume. It is worth noting that the backlit system has a very small optical power density where the scene volume is $4.8\,\text{m}^3$ and total light source power output is $0.88\,\text{W}$ (for two LEDs), thus, light power density is $0.18\,\text{W m}^{-3}$. The new system provides slightly larger scene volume (due to greater height of the Fresnel lenses) of $5.9\,\text{m}^3$, but light source power output is approximately $12\,\text{W}$ (for two LED ring lights) and, thus, the optical power density is $2\,\text{W m}^{-3}$. Repeat experiments and rotation of the human bait show that the increased power density from the fixed position of the illumination does not influence the behaviour of the mosquitoes. There are additional benefits in uniformity of the illumination into the corners of the measurement volume compared to the backlit set-up as evidenced by the spatial distribution data in figures 5 and 6.

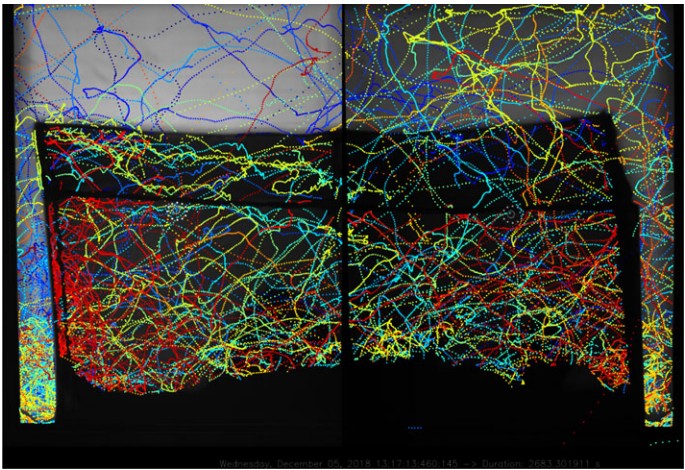

**Figure 8.** Example of full RRS system segmented recordings using Ximea CB120RG-CM cameras and 5 ms exposure per frame. Left camera, 0 dB gain; right camera, −3.5 dB gain.

By contrast, backlit illumination causes a hot spot in the middle of the image and any increase in source power will cause saturation and prevent mosquito detection. The more uniform illumination from RRS imaging, on the other hand, allows the light level to be raised significantly without causing saturation of the image in the centre.

The 12 LEDs in the ring light are necessarily displaced from the camera lens axis and therefore create collimated beams that traverse the measurement volume at a small angle, less than 4°, to the optical axis of the Fresnel lens. This reduces any shadowing caused by other objects, such as the bednet, affecting the image of the mosquitoes. The position and size of the diffuser in the RRS system (being the full aperture RRS itself) gives increased scattering compared to that in the backlit set-up. The scatter from the RRS having a larger angular subtense than the propagation directions of the illumination gives a diffuse cone of illumination of greater than 10°. These effects are probably key to giving increased mosquito segmentation performance over the critical bednet areas, leading to the level of mosquito detection across the bednet for RRS of 92% compared to 54% for backlit imaging (table 1). The practical benefits are important for ease of use in field settings, e.g. in Banfora, Burkina Faso. The RRS approach requires two Fresnel lenses (rather than four for the backlit case) and two tripods for the camera-LED assemblies (four tripods for two cameras and two light sources for backlit). The RRS itself can be assembled in the field from locally sourced plywood sheets and the retro-reflective tape bonded to it.

The performance of the RRS approach to track mosquitoes in front of and behind the bednet has been evaluated in laboratory experiments where the mosquitoes are released via an aspirator at the specific locations. Segmented and tracked results are given in figure 7a when the mosquitoes are released in front of the bednet and in figure 7b where the point of release is behind the bednet. The image of the aspirator at the moment of release is clearly visible on the right edge of each image. Several mosquitoes were released simultaneously in each case. It is clear that mosquitoes located closer to the Fresnel lens (in front of the net) are reliably identified by the segmentation algorithm everywhere—the tracks show a regular spacing of segmented positions. Mosquitoes flying closer to the retro-reflective screen (behind the net) are consistently identified above the net and across the vertical net walls. Segmentation performance is slightly worse across the inclined net roof, although most positions are still identified and sufficient to enable flight path tracking. This occurs due to the combination of limited depth of field and additional optical attenuation through the inclined net which gives increased occlusion. Also, at the beginning of the tracks mosquitoes were not identified well because they were blown from the aspirator quite fast (much faster than their natural maximum velocity).

Figure 8 shows identified mosquito positions from both cameras with a human volunteer lying under the net. Qualitative inspection reveals a high level of activity in front of and behind the bednet as well as bouncing behaviours particularly evident above the feet of the person (left-hand side but with some continuity to the right-hand camera view). These data were obtained with Ximea CB120RG-CM cameras. The left-hand camera used 5 ms exposure time with 0 dB gain, whereas the right-hand camera was used with 5 ms exposure and −3.5 dB gain. The continuity of tracks between the two views reflects the robustness of the segmentation algorithm to these varying conditions. Movement of the volunteer and draughts cause movement of the bednet and the associated strings supporting it

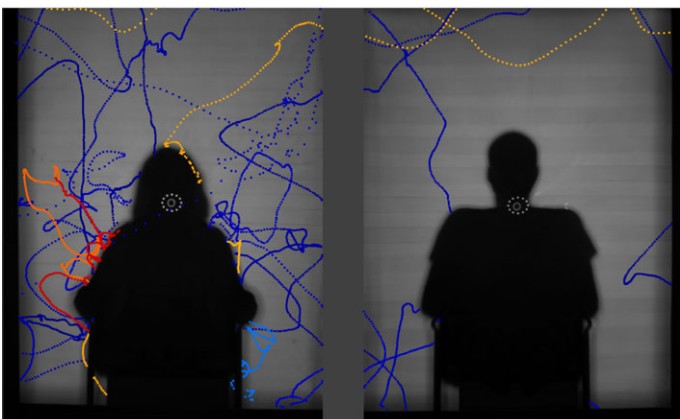

**Figure 9.** Exemplar image captured using the RRS recording system described herein, showing 10 min flight activity by 10 *Anopheles gambiae s.l.* females around two seated adults. Higher flight activity is apparent at the person on the left, who was bitten twice while the individual on the right received three bites.

and lead to some false segmentation in individual frames that are largely filtered out via the tracking algorithms. Electronic supplementary material, video file demonstrates short part of the recording, where mosquitoes move from one side to the other and some bouncing activity on the roof of the net.

The quantitative performance metrics presented in table 1 are from different experiments for the backlit and RRS imaging systems and hence exhibit natural variability due to the use of different mosquitoes and human bait. This is evident in the detected positions in tracks metric, which shows that the RRS experiment produced a higher number of segmented mosquito positions that passed the consistency requirements to be part of mosquito flight tracks. The data on gap size in tracks show that the RRS imaging approach gave similar performance both across net and outside net regions. By contrast, the backlit imaging set-up gives larger track gaps across the bednet probably due to reduced illumination levels away from the centre of each camera's field of view and the reduced transmission of the bednet in this region. The variability in gap size (given by the standard deviation, s.d.) is higher for both regions with backlit imaging, but especially so for the across net region. The data for the number of track gaps normalized by the number of detected positions show similar trends: consistent performance from the RRS set-up in both across net and outside net regions and worse metrics with backlit imaging.

The track length data show that average performance of backlit and RRS imaging systems is similar. However, the uninterrupted track length is significantly greater for RRS than backlit, indicating the higher consistency in segmenting mosquito positions. The mosquito detection percentages show that the RRS imaging approach achieves consistent performance in both across net and outside net regions with detection levels greater than 90% that are significantly higher than for backlit imaging. It is worth noting again that the same processing pipeline was applied to the recordings made with backlit and RRS recording systems. So, the differences are due to the optical set-up rather than signal processing. With this improved performance in mosquito image segmentation, the tracking algorithms are able to reliably bridge the gaps in tracks that do occur and hence deliver significantly improved flight tracks for entomologists.

The global threat to public health from mosquito-borne diseases like malaria, dengue, yellow fever and emerging arboviral infections such as Zika, together with the spread of insecticide resistance, demonstrate the unabated challenge for mosquito control and the relentless need for novel control tools. Tracking has already proven its value for advancing bednet design [4], but the performance of the RRS imaging approach has broad potential for numerous studies, ranging from basic research to evaluation of novel control tools. A widely applicable example is the tracking of mosquitoes or other flying insects during approach and landing on single or multiple test targets. Binary choice-tests are used widely to measure the effects of repellents or other attractants on mosquitoes, or genetic or phenotypic differences between mosquito populations or laboratory strains. Figure 9 illustrates the potential of the RRS system to generate images suitable for a typical study comparing attractiveness to mosquitoes of two human subjects, e.g. to quantify the degree to which a repellent treatment on one host kills or possibly diverts mosquitoes to the untreated hosts.

Finally, the more compact and easily aligned RRS system greatly facilitates transportation between and installation at multiple sites, including rural or relatively remote sites. Multiple RRS tracking

systems are now being installed in east and west Africa where they will be used to characterize the efficacy and monitor the progress of the so-called next generation of insecticidal bednets; essential evidence for decision-makers planning effective malaria control of pyrethroid-resistant mosquitoes.

# 4. Conclusion

The introduction of a RRS to large-volume, human-baited mosquito tracking studies delivers significant improvements to mosquito detection rates in comparison to previous backlit imaging approaches. This is primarily due to increased uniformity of illumination and the narrow-angle diffuse nature of the reflected light from the RRS. A disadvantage of RRS imaging is that the light double passes each bednet layer that is typically (although, not necessarily) the focus of the experiments, reducing the available light intensity. However, the use of large-kernel Gaussian noise reduction filters combined with low detection thresholds of 1 to 2 greyscales have been shown to give robust mosquito detection. With RRS imaging, mosquito detection rates of 92% have been obtained, compared to an average of 69% with backlit imaging.

There are further practical benefits of the RRS imaging set-up in terms of reduced footprint, reduced number of optical components required and reduced sensitivity to mis-alignment due to the retro-reflective nature of the material.

RRS-based imaging set-ups are a powerful tool in entomological research, especially for species like the *Anopheles* mosquitoes that transmit malaria, nocturnally active mosquitoes that require remote detection as their natural behaviour is influenced by the presence of a human observer. RRS systems are installed in research laboratories in the UK and Burkina Faso, where the insight they provided on malaria vector behaviour led directly to novel next-generation bednet designs and an innovative solution to the threat from insecticide resistance [4].

Ethics. All research methods were performed in accordance with approved guidelines for those procedures and written informed consent was obtained from all volunteer subjects. Video recording work was approved by the Research Ethics Committees at the Liverpool School of Tropical Medicine (LSTM Research Protocol 16-38, 11 October 2016, Liverpool) and Centre National de Recherche et de Formation sur le Paludisme (CNRFP Deliberation no. 2016-9-097, 20 September 2016, Ouagadougou). No adverse effects of treatment or mosquito-borne infections were reported by volunteers during the course of the study.

Data accessibility. Image segmentation software source code is available at [29], all other custom codes and all data used in this work are available at [30] and larger datasets used for illumination analysis are available at [31].

Authors' contributions. C.K., D.P.T., C.E.T. and V.V. conceived the idea of the improved system. V.V., C.K. and C.S. developed the new system and did initial testing. P.J.M. and G.P.D.M. gave advice on specifics of entomological use and field conditions for the system. G.P.D.M. and V.V. set up both field and laboratory-based systems and did initial testing of the system at the field site. G.P.D.M. collected field test data, whereas A.G. collected test data from laboratory-based system. C.K., C.S. and V.V. did initial pre-processing and initial assessment of the collected data. V.V. together with C.K., C.E.T. and D.P.T. developed and implemented assessment metrics and methods. C.K. with the help of C.S., C.E.T., D.P.T. and V.V. devised and implemented improvements to the segmentation algorithm. V.V. with the help of C.K. and C.S. ran the performance analysis. V.V., C.K., C.E.T and D.P.T. wrote the paper with contributions from P.J.M. All authors approved the final submitted version.

Competing interests. The authors declare no competing interests.

Funding. This research was supported with funding from the UK Medical Research Council (MRC) and the UK Department for International Development (DFID) under the MRC/DFID Concordat agreement (MR/M011941/1), a Wellcome Trust Collaborative Award (Malaria in Insecticide Resistant Africa MiRA; 200222/Z/15/Z) and the Bill & Melinda Gates Foundation (OPP1159078).

Acknowledgements. The authors thank the volunteer sleepers both in the laboratory in Liverpool and in the field in Burkina Faso and are especially grateful to the community at Tengrela, Burkina Faso, who continue to support studies at their village. Also, the authors are very grateful for all constructive comments from the reviewers, which greatly helped to improve the paper.

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
