## [Reviewer comments · Royal Society Open Science]

Review History

RSOS-191951.R0 (Original submission)

Review form: Reviewer 1

Is the manuscript scientifically sound in its present form?

Yes

Are the interpretations and conclusions justified by the results?

Yes

Is the language acceptable?

Yes

Do you have any ethical concerns with this paper?

No

Have you any concerns about statistical analyses in this paper?

No

Recommendation?

Major revision is needed (please make suggestions in comments)

Comments to the Author(s)

Reducing transmission of mosquito-borne illness requires simple and robust methods to quantify behavior in natural settings. An example is measuring flight trajectories around a bednet covering an individual, but current tracking methods suffer from non-uniform illumination, which complicates automated tracking due to changing background light levels. Here the authors introduce a method for providing much more uniform illumination via a retro-reflective lens behind the bednet. This substantially improves tracking, in particular enabling longer continuous detection of flight trajectories, which could benefit quantitative behavioral analysis.

This work provides a useful methodological advance in collecting insect flight data and could improve robustness of data collection and analysis in conditions outside the laboratory. The technical language is precise, which will enable others to easily reproduce the setup.

The major addition that would improve the scientific value of the paper would be a demonstration of the method to yield a novel scientific finding. One idea would be to expand on the “attractiveness” question exemplified in Figure 9. What makes one person more attractive to mosquitoes than another is important to understand, and it would be interesting to see quantitatively how the retro-reflective screen method improves one’s ability to assess differences in attractiveness. It would also be interesting to use this method, along with more detailed behavioral analysis, to quantify distributions of mosquito behaviors and see how they changed in different conditions. E.g. which behaviors do mosquitoes engage in more frequently when in the presence of an attractive host? Or which behavioral patterns differentiated disease-carrying vs non-disease-carrying mosquitoes?

Minor notes/questions:

1. The technical writing was precise but a bit verbose. E.g. “Large field of view back lit imaging systems have been reported with two parallel imaging channels to give a measurement volume of $2 \times 2 \times 1.4$ m in total with large aperture Fresnel lenses enabling collimated illumination and telecentric imaging” is a difficult first sentence of a paragraph to parse. It could improve the manuscript to put more emphasis up front on the core advance of using a retro-reflective screen to create more uniformly distributed light and allocate details to the methods section or an appendix.
2. On P2 L35, it’s stated that mosquito images are occluded when the mosquito is in front of the bednet. I don’t understand why occlusion would occur if the bednet is behind the mosquito.
3. On P3 L24, the phrase “per camera” is used, suggesting there are multiple cameras in this setup. However, it was my understanding that there is only one camera involved when the retro-reflective screen method is used.
4. Would the illumination level used in this method be expected to affect mosquito behavior at all? Or is it far enough in the IR spectrum as not interfere?
5. The authors mention that a detection threshold must be set manually (P6 L48). Would this need to be done repeatedly as ambient light levels change throughout the night?
6. If possible, it would be useful to compare the RRS system to standard backlit systems across light levels. For example, can a backlit system be just as good as an RRS system when there is more light? Or do e.g. saturation effects preclude this?
7. It would be useful to say a bit more about whether the tracking algorithm loses the mosquito when it lands for a few seconds. I imagine this algorithm would indeed lose the mosquito, since it operates on frame-to-frame differences.

8. Is it possible at all to track the mosquito in the darkest region of the scene (i.e. in front of the volunteer)?

9. I understand the practicality of having simple tracking algorithms for deployment in the field, but it would be good nonetheless to provide more comparison of this algorithm to state-of-the-art algorithms used in other insect flight-tracking studies. For example, see Straw, Andrew D., et al. "Multi-camera real-time three-dimensional tracking of multiple flying animals." *Journal of The Royal Society Interface* 8.56 (2010): 395-409.

Review form: Reviewer 2

Is the manuscript scientifically sound in its present form?

Yes

Are the interpretations and conclusions justified by the results?

Yes

Is the language acceptable?

Yes

Do you have any ethical concerns with this paper?

No

Have you any concerns about statistical analyses in this paper?

No

Recommendation?

Accept with minor revision (please list in comments)

Comments to the Author(s)

This is a revised version of the original paper. I have the following concerns :

- Major point : I do not see clearly the difference between the RRS system and the backlit one. Authors must provide a figure (like figure 1) which shows clearly the two setup.

Minor points:

- model of the fresnel lens also supplier must be indicated
- were telecentric objectives used for the camera ? It is not clear. If so, please indicate the model and the optical parameter of the objectives used in this study.
- how big was the surface covered by the tape of 3M material ? Please indicate the cost ?
- authors must provide the code used for the image processing. The code must be made available on a GitHub for example.
- the attached video must be provided with explanations (comments).

Review form: Reviewer 3

Is the manuscript scientifically sound in its present form?

Yes

Are the interpretations and conclusions justified by the results?

Yes

Is the language acceptable?

Yes

Do you have any ethical concerns with this paper?

No

Have you any concerns about statistical analyses in this paper?

No

Recommendation?

Accept with minor revision (please list in comments)

Comments to the Author(s)

The authors provide details of an improved technique to track free-flying nocturnal mosquitoes around bed nets in 2D. Although a highly specific, context dependent, research set-up, the technique is useful to answer fundamental questions about mosquito behavior around bed nets or can be used for some other vector arthropods. The manuscript is well written.

The manuscript is an 'improved-methods' paper, mainly by adding a retro-reflective screen to improve illumination conditions. Figure 9 in particular is an exemplar image of how the RRS system can be implemented, however it does not provide the analysed data. The authors focus on the relevant 2D data they obtained earlier, using a previous version of the system, but seem to ignore other research/techniques in the lab or semi-field where mosquitoes were successfully tracked under challenging conditions and where flight paths were reconstructed in 3D. The introduction (p4) and proposed methods focuses on back-lit tracking and offline acquisition. There are good reasons for choosing this method, but the authors could highlight, or discuss, that other illumination methods are possible too and that online tracking would tackle their challenge with processing large data files and data handling.

I concur with many of the original comments made by reviewer #1. I understand from the authors' reply that the improved method (adding RRS) was already implemented during the studies of the recently published ref.4. This would indeed be an argument to have the updated, detailed methods, included within that paper, rather than publishing a separate methods paper.

Additional minor comments/suggestions:

- p3, line 25: most? There are many more mosquito- behavioural activities to quantify I can think of
- Reference #4 is published now right? (remove accepted, complete reference)
- p5, line 23: tracking ..
- p10, line 38: spelling coluzzii (no capital, double i)

Decision letter (RSOS-191951.R0)

17-Jan-2020

Dear Dr Voloshin,

The editors assigned to your paper ("Diffuse retro-reflective imaging for improved video tracking of mosquitoes at human baited bednets") have now received comments from reviewers. We would like you to revise your paper in accordance with the referee and Associate Editor suggestions which can be found below (not including confidential reports to the Editor). Please note this decision does not guarantee eventual acceptance.

Please submit a copy of your revised paper before 09-Feb-2020. Please note that the revision deadline will expire at 00.00am on this date. If we do not hear from you within this time then it will be assumed that the paper has been withdrawn. In exceptional circumstances, extensions may be possible if agreed with the Editorial Office in advance. We do not allow multiple rounds of revision so we urge you to make every effort to fully address all of the comments at this stage. If deemed necessary by the Editors, your manuscript will be sent back to one or more of the original reviewers for assessment. If the original reviewers are not available, we may invite new reviewers.

- Data accessibility

If you wish to submit your supporting data or code to Dryad (<http://datadryad.org/>), or modify your current submission to dryad, please use the following link:
<http://datadryad.org/submit?journalID=RSOS&manu=RSOS-191951>

- Competing interests

- Authors' contributions

- Acknowledgements

- Funding statement

Kind regards,

Andrew Dunn

on behalf of Prof R. Kerry Rowe (Subject Editor)

Associate Editor's comments:

We have now received 3 reviewer reports on your manuscript. This paper was previously reviewed at Interface, but unfortunately neither of the two original Interface referees were available to comment on the revisions made in the transferred paper. Owing to the concerns raised by Referee 1 (and the additional comments outlined by Reviewers 2 and 3), we ask that you please carefully revise your manuscript. We appreciate that this may take some time, so please just let us know if you require additional time to revise your manuscript. In your revised submission, please supply a clean copy of your paper as well as a tracked edits; highlighting the revisions you have made. Thanks for your submission to RSOS, and we look forward to your revision.

Comments to Author:

Reviewers' Comments to Author:

Reviewer: 1

Comments to the Author(s)

Reducing transmission of mosquito-borne illness requires simple and robust methods to quantify behavior in natural settings. An example is measuring flight trajectories around a bednet covering an individual, but current tracking methods suffer from non-uniform illumination, which complicates automated tracking due to changing background light levels. Here the authors introduce a method for providing much more uniform illumination via a retro-reflective lens behind the bednet. This substantially improves tracking, in particular enabling longer continuous detection of flight trajectories, which could benefit quantitative behavioral analysis.

This work provides a useful methodological advance in collecting insect flight data and could improve robustness of data collection and analysis in conditions outside the laboratory. The technical language is precise, which will enable others to easily reproduce the setup.

The major addition that would improve the scientific value of the paper would be a demonstration of the method to yield a novel scientific finding. One idea would be to expand on the “attractiveness” question exemplified in Figure 9. What makes one person more attractive to mosquitoes than another is important to understand, and it would be interesting to see quantitatively how the retro-reflective screen method improves one’s ability to assess differences in attractiveness. It would also be interesting to use this method, along with more detailed behavioral analysis, to quantify distributions of mosquito behaviors and see how they changed in different conditions. E.g. which behaviors do mosquitoes engage in more frequently when in the presence of an attractive host? Or which behavioral patterns differentiated disease-carrying vs non-disease-carrying mosquitoes?

Minor notes/questions:

1. The technical writing was precise but a bit verbose. E.g. “Large field of view back lit imaging systems have been reported with two parallel imaging channels to give a measurement volume of 2 x 2 x 1.4 m in total with large aperture Fresnel lenses enabling collimated illumination and telecentric imaging” is a difficult first sentence of a paragraph to parse. It could improve the manuscript to put more emphasis up front on the core advance of using a retro-reflective screen to create more uniformly distributed light and allocate details to the methods section or an appendix.
2. On P2 L35, it’s stated that mosquito images are occluded when the mosquito is in front of the bednet. I don’t understand why occlusion would occur if the bednet is behind the mosquito.
3. On P3 L24, the phrase “per camera” is used, suggesting there are multiple cameras in this setup. However, it was my understanding that there is only one camera involved when the retro-reflective screen method is used.
4. Would the illumination level used in this method be expected to affect mosquito behavior at all? Or is it far enough in the IR spectrum as not interfere?
5. The authors mention that a detection threshold must be set manually (P6 L48). Would this need to be done repeatedly as ambient light levels change throughout the night?
6. If possible, it would be useful to compare the RRS system to standard backlit systems across light levels. For example, can a backlit system be just as good as an RRS system when there is more light? Or do e.g. saturation effects preclude this?
7. It would be useful to say a bit more about whether the tracking algorithm loses the mosquito when it lands for a few seconds. I imagine this algorithm would indeed lose the mosquito, since it operates on frame-to-frame differences.
8. Is it possible at all to track the mosquito in the darkest region of the scene (i.e. in front of the volunteer)?
9. I understand the practicality of having simple tracking algorithms for deployment in the field, but it would be good nonetheless to provide more comparison of this algorithm to state-of-the-art algorithms used in other insect flight-tracking studies. For example, see Straw, Andrew D., et al. "Multi-camera real-time three-dimensional tracking of multiple flying animals." *Journal of The Royal Society Interface* 8.56 (2010): 395-409.

Reviewer: 2

Comments to the Author(s)

This is a revised version of the original paper. I have the following concerns :

- Major point : I do not see clearly the difference between the RRS system and the backlit one. Authors must provide a figure (like figure 1) which shows clearly the two setup.

Minor points:

- model of the fresnel lens also supplier must be indicated
- were telecentric objectives used for the camera ? It is not clear. If so, please indicate the model and the optical parameter of the objectives used in this study.
- how big was the surface covered by the tape of 3M material ? Please indicate the cost ?
- authors must provide the code used for the image processing. The code must be made available on a GitHub for example.
- the attached video must be provided with explanations (comments).

Reviewer: 3

Comments to the Author(s)

The authors provide details of an improved technique to track free-flying nocturnal mosquitoes around bed nets in 2D. Although a highly specific, context dependent, research set-up, the technique is useful to answer fundamental questions about mosquito behavior around bed nets or can be used for some other vector arthropods. The manuscript is well written.

The manuscript is an 'improved-methods' paper, mainly by adding a retro-reflective screen to improve illumination conditions. Figure 9 in particular is an exemplar image of how the RRS system can be implemented, however it does not provide the analysed data. The authors focus on the relevant 2D data they obtained earlier, using a previous version of the system, but seem to ignore other research/techniques in the lab or semi-field where mosquitoes were successfully tracked under challenging conditions and where flight paths were reconstructed in 3D.

The introduction (p4) and proposed methods focuses on back-lit tracking and offline acquisition. There are good reasons for choosing this method, but the authors could highlight, or discuss, that other illumination methods are possible too and that online tracking would tackle their challenge with processing large data files and data handling.

I concur with many of the original comments made by reviewer #1. I understand from the authors' reply that the improved method (adding RRS) was already implemented during the studies of the recently published ref.4. This would indeed be an argument to have the updated, detailed methods, included within that paper, rather than publishing a separate methods paper.

Additional minor comments/suggestions:

- p3, line 25: most? There are many more mosquito- behavioural activities to quantify I can think of
- Reference #4 is published now right? (remove accepted, complete reference)
- p5, line 23: tracking ..
- p10, line 38: spelling coluzzii (no capital, double i)

Author's Response to Decision Letter for (RSOS-191951.R0)

See Appendix A.

RSOS-191951.R1 (Revision)

Review form: Reviewer 1

Is the manuscript scientifically sound in its present form?

Yes

Are the interpretations and conclusions justified by the results?

Yes

Is the language acceptable?

Yes

Do you have any ethical concerns with this paper?

No

Have you any concerns about statistical analyses in this paper?

No

Recommendation?

Reject

Comments to the Author(s)

Thank you for your responses to the reviewer comments. They have helped clarify and contextualize the work, and the work appears technically sound. Unfortunately, while I agree that the new RRS system represents a useful technique for improving mosquito flight tracking in field conditions, since the authors don't provide a strong new scientific result arising from their methodology, I don't believe this is an appropriate venue for publication. I think it would make more sense to publish this as part of the methods section for the ongoing experimental work the authors mentioned in their response.

Decision letter (RSOS-191951.R1)

06-Apr-2020

Dear Dr Voloshin,

It is a pleasure to accept your manuscript entitled "Diffuse retro-reflective imaging for improved video tracking of mosquitoes at human baited bednets" in its current form for publication in Royal Society Open Science. The comments of the reviewer(s) who reviewed your manuscript are included at the foot of this letter.

You can expect to receive a proof of your article in the near future. Please contact the editorial office (openscience_proofs@royalsociety.org) and the production office (openscience@royalsociety.org) to let us know if you are likely to be away from e-mail contact -- if

you are going to be away, please nominate a co-author (if available) to manage the proofing process, and ensure they are copied into your email to the journal.

Kind regards,

Lianne Parkhouse
Royal Society Open Science
openscience@royalsociety.org

on behalf of the Associate Editor, and Professor R. Kerry Rowe (Subject Editor)
openscience@royalsociety.org

Associate Editor Comments to Author:

The Editors have been presented with a tricky decision: on the one hand, the most critical reviewer from the first round of review recommends rejection, and on the other, they note that the work is technically sound. Given that Royal Society Open Science avoids making decisions on impact or importance (unless a manuscript is trivial), we have made the decision to accept the paper for the following reasons:

1. As above, the journal makes decisions to publish on the grounds that a paper must be technically sound - impact/importance is not routinely considered. The reviewer here notes your paper is technically sound.
2. The reviewer also notes that you responded effectively to their criticisms. This is important, and without your engagement in the process it would not have provide important context for a reader.
3. As the reviewers in the prior iteration were broadly in favour of publication, it would not appear to be equitable for the paper to be rejected at this stage.

That said, we would recommend that you take onboard the reviewer's feedback regarding the structuring of the work: it would, in general, be better to prepare one, more complete paper, than find yourself in the situation of preparing several papers that only tell part of the broader story you are aiming to deliver. Nevertheless, thanks for your support of Royal Society Open Science.

Reviewer comments to Author:

Reviewer: 1
Comments to the Author(s)

Thank you for your responses to the reviewer comments. They have helped clarify and contextualize the work, and the work appears technically sound. Unfortunately, while I agree that the new RRS system represents a useful technique for improving mosquito flight tracking in field conditions, since the authors don't provide a strong new scientific result arising from their

methodology, I don't believe this is an appropriate venue for publication. I think it would make more sense to publish this as part of the methods section for the ongoing experimental work the authors mentioned in their response.

Appendix A

ROYAL SOCIETY OPEN SCIENCE

rsos.royalsocietypublishing.org

Research

Article submitted to journal

Subject Areas:

Optics Engineering, Vector Biology

Keywords:

mosquito tracking, retro-reflective screen, entomology

Author for correspondence:

Vitaly Voloshin

e-mail: v.voloshin@warwick.ac.uk

Diffuse retro-reflective imaging for improved video tracking of mosquitoes at human baited bednets

Vitaly Voloshin¹, Christian Kröner¹, Chandrabhan Seniya¹, Gregory P. D. Murray², Amy Guy², Catherine E. Towers¹, Philip J. McCall², David P. Towers¹

¹School of Engineering, University of Warwick, Coventry, CV4 7AL, UK

²Liverpool School of Tropical Medicine, Pembroke Place, Liverpool, L3 5QA, UK

Robust imaging techniques for tracking insects have been essential tools in numerous laboratory and field studies on pests, beneficial insects and model systems. Recent innovations in optical imaging systems and associated signal processing have enabled detailed characterisation of nocturnal mosquito behaviour around bednets and improvements in bednet design, a global essential for protecting populations against malaria. Nonetheless, there remain challenges around ease of use for large scale *in situ* recordings and extracting data reliably in the critical areas of the bednet where the optical signal is attenuated. Here we introduce a retro-reflective screen at the back of the measurement volume, which can simultaneously provide diffuse illumination, and remove optical alignment issues whilst requiring only one-sided access to the measurement space. The illumination becomes significantly more uniform, although, noise removal algorithms are needed to reduce the effects of shot noise particularly across low intensity bednet regions. By systematically introducing mosquitoes in front and behind the bednet in lab experiments we are able to demonstrate robust tracking in these challenging areas. Overall, the retro-reflective imaging setup delivers mosquito segmentation rates in excess of 90% compared to less than 70% with back-lit systems.

1. Introduction

Many arthropod vectors of human infections are highly adapted to the human home and in many regions worldwide, transmission of malaria, leishmaniasis, Chagas disease, lymphatic filariasis and tick-borne relapsing fever occurs when the vectors take blood from sleeping humans. Bednets treated with insecticide can be very effective in preventing transmission of these infections and in Africa, the factory-treated durable type of nets referred to as long-lasting insecticidal nets (LLINs), are the most effective method available and an essential element of malaria control and elimination strategies today. Increased understanding of vector behaviour at LLINs, including how different LLIN treatments affect mosquito flight around and contact with the net surface is essential for understanding LLIN modes of action [1, 2] and raises the prospect of developing more effective interventions to reduce disease transmission [3, 4].

Optical imaging techniques have been used for decades in entomological studies in diverse settings in the laboratory and field, and recently a tracking system was used to characterise in detail how *Anopheles gambiae* mosquitoes, the vectors of human malaria in sub-Saharan Africa, interact with human-occupied bednet [1, 2, 5]. Offering high spatial and temporal resolution, optical imaging has clear advantages for quantifying vector behaviour [6, 7], but their application in typical sub-Saharan Africa dwellings present particular challenges. ~~In order to~~ In Africa, insecticidal bednets are by far the most effective intervention available for malaria control and the most widespread method used to prevent its transmission [8–10]. Sustaining the required high level of efficacy against increasingly insecticide resistant vector populations requires novel insecticide treatments on bednets, and vector biologists need to understand better how hungry host-seeking mosquitoes ~~are affected by~~ interact with different treatments ~~the bednet~~, or how net alterations (physical as well as chemical) might ~~improve net performance,~~ affect efficacy. ~~entomologists are most~~ To investigate this, they are particularly interested in visualizing activity around the sleeping ~~space~~ human body, the bednet suspended above it and the regions around this in order to examine details of approach, attack and departure [11, 12]. Hence, the inspected volume ideally needs to be 2 × 2.5 × 1.5 m (depth × width × height), which generously encompasses the space around a typical installed bednet. To avoid rapidly varying spatial resolution between the front and back of the measurement volume, telecentric approaches are desirable for both illumination and imaging and given the space constraints in sub-Saharan dwellings this leads to typically 0.5 Numerical Aperture (NA) optical systems. The bednet itself is a regular grid of polyester or polyethylene fibre (typically 75–180 denier, mesh size of 24–32 holes / cm²) that can easily occlude the images of a slender unfed *Anopheles gambiae* female (body length and thickness typically less than 10 mm and 3 mm, respectively) [13, 14]. ~~that occludes the mosquito images – particularly as multiple layers of netting are often used. The mesh size in the bednet is typically smaller than the mosquito,~~ *Anopheles gambiae* are 2.5–3.5 mm in wing length, ~~h~~ Hence the visibility (contrast) of the mosquito's images ~~are partially occluded~~ is reduced when the mosquito is in front of, or behind a bednet as the bright background field is attenuated from transmission through the bednet layers each side of the human bait. Furthermore, for field studies the optical system needs to be simple to ~~setup~~ transport and install and sufficiently robust to ~~withstand~~ environmental instabilities, e.g. flexible wooden floors, as well as computationally efficient to extract the flight tracks required by entomologists.

Two and three dimensional (multi camera) imaging setups have been reported using illuminated diffuse surfaces or lamps as a background [15, 16]. Combined with algorithms for tracking, two or three dimensional flight trajectories are produced from which responses to attractants or interventions can be determined via manual inspection. However, such analyses whilst of value, ~~do~~ cannot determine responses to an insecticide treated bednet ~~because of the inability to capture large enough volume, which remains the most widespread and effective intervention against disease transmission, or with native mosquito populations.~~ Field studies to examine mating behaviour have been reported using the setting sun as a back light with a pair of stereo cameras to give 3-D mosquito tracks over volumes of metre-scale dimensions but

this imaging approach cannot be translated to the inside of dwellings in nocturnal situations [17–19]. Stereo or multi-camera 3D imaging provides spatial resolution that increases proportionately with the field of view [20, 21]. Another multi-camera approach (up to 11 cameras) for 3D tracking flying animals was reported in [22]. The tracking in [22] relies on high levels of spatial sampling in order to extract position and orientation information of the animal as well as on the use of a network of processing computers (up to 9 computers).

Millimetre scale resolution should be available over room size volumes from optically suitable surfaces, but performance will degrade with mosquito targets that vary in presentation according to angle of view of the multiple cameras. Furthermore, a minimum of two camera views are needed in each region for 3D metrology, hence to adequately map the space around a human baited bednet would require pairs of cameras for each side, the head and feet areas as well as cameras to map the space above the bednet. The entirety of the bednet surface needs to be captured as this is where mosquitoes interact with the insecticide. Hence, the test room would need to have 5 sides largely transparent in order to position the cameras outside of the room and look in (meaning modifications to the roof region where mosquitoes are known to enter via eaves), or a significantly larger sized building that would be atypical compared to sub-Saharan dwellings. Moreover, multi-camera 3D systems requires significant levels of processing power that is often not available in the field.

Large field of view back lit imaging systems which record the whole bed with bednet and surrounding areas have been reported [5]. The setup uses with two parallel imaging channels (two non-intersecting camera images) to give a measurement volume of $2 \times 2 \times 1.4$ m in total. Large aperture Fresnel lenses enabling collimated illumination and telecentric imaging. Illumination was provided by a single LED with a transmission diffuser located behind the back Fresnel lens for each camera. As part of this study, algorithms were also reported that produced flight tracks of 25 mosquitoes over hour-long recording periods of up to 2 hours. The recording system enabled discrimination of four behavioural modes were identified during the mosquito's interactions with a human-occupied bednet: non-contact flights (swooping), and flights with single (visiting) (the bednet), multiple-rapid (bouncing) and/or sustained (resting) net contact [5]. The data gave This became the basis for elucidation of insight into the mode of action of insecticidal bednets [8], studies that, in turn, led to mosquito host-seeking behaviour and quantified an individual mosquito's contact with the bednet (and hence insecticide) with the information leading to the 'barrier bednet', an innovative and field-tested concept that could greatly expand the potential and lifespan of insecticidal bednets [4, 23]. Despite these advances, conducting these experiments in the field is challenging. The imaging approach worked in transmission with a Fresnel lens at either end of the measurement volume, consequently 1.5 m was needed beyond the lenses at each end. The use of two Fresnel lenses per camera also generates undesirable amplitude modulation in the images in circular rings and needs very careful alignment of the two lenses with respect to each other. The unstable nature of test environments in sub-Saharan Africa dwellings means that the alignment of these large aperture Fresnel lenses needed regular adjustment.

Here we introduce an optical setup using a retro-reflective screen (RRS) to eliminate the optical alignment problems and, reduce the size of the optical system and improve the uniformity of the images. In this approach a single Fresnel lens per camera is used per camera to both collimate the illumination beam and focus the light reflected from the RRS at the far end of the measurement volume. No further optical components are needed beyond the RRS. Similarly to the backlit approach, the volume captured is $2 \times 2 \times 1.4$ m and two parallel imaging channels are used with one camera per channel. In transmission through a bednet, the light amplitude is reduced and in this reflective mode the light is attenuated through twice the number of bednet layers compared to the back-lit setup. Additional data processing steps are introduced to handle the increased range of contrast in the images, in particular, to manage the reduced contrast of the mosquito images partially obscured by layers of net. The following sections describe the optical setup and signal processing. Sets of experimental data are shown where mosquitoes are introduced into known spatial locations, e.g. in front or behind a bednet – as a means of confirming mosquito detection

ability in all regions of the image. Hence, tracking performance comparisons are made between the back-lit and RRS based imaging approaches.

Finally, Beyond bednets that provide protection to sleeping hosts, variations in chemosensory cues including carbon dioxide and volatile organic compounds from the skin are widely recognised as determining the attractiveness of individual people. Assessment of variation in individual attractiveness to a host-seeking mosquito is an important first step in being able to assess novel interventions, e.g. insecticide treated clothing or topical repellents that can offer protection away from a sleeping space. Even though the work is in its early stages, the RRS based tracking system's potential Initial mosquito tracks that are shown as for a range of behaviour studies is shown by its performance in a binary test of the relative attractiveness between two individuals of two human hosts, where multiple mosquitoes are tracked as they orient to and select the individual on whom they will land and bloodfed.

2. Methods

(a) Optical Setup

The optical setup for the RRS approach is shown in Figure 1b and for comparison purposes, the backlit setup is in Figure 1a. Light from an LED ring light expands over the Fresnel lens (approx. 1.4 x 1.0 m aperture, 1.2 m focal length, NTKJ CO., LTD. CF1200 [24]) and is then approximately collimated to illuminate the space above the bednet and through the bednet itself. A volunteer acts as human bait lying beneath the bednet. The telecentric setup (created by the combination of camera and Fresnel lenses) plays a key role in image formation and metrology of mosquito position. As the depth of the scene is quite large, approximately 2 metres, telecentric imaging is essential to determine the mosquito displacements accurately when located at different depths. Moreover, the collimated illumination and imaging enables neighbouring camera views to be independent and give sufficient spatial resolution over the measurement volume.

The light travels to a back wall where it is reflected back via a RRS. Whilst a number of retro-reflective materials are available, improved performance was found using 3M™ Scotchlite™ High Gain 7610 [25] obtained as a tape to be stuck to the back wall. The material contains glass beads and is based on an exposed lens material, which means it does not have a glossy surface covering but a matte appearance producing a diffuse background with scattered light over a 10° reflection cone. The RRS retroreflective material applied to a plywood board (size 2.4 m x 1.2 m) per camera is placed approximately 2 m from the Fresnel lens. The reflected light re-crosses the measurement volume and is focused by the same Fresnel lens which forms a telecentric lens pair with an imaging optic mounted on the camera. This configuration allows illumination and imaging from one side of the bednet and scene and is relatively insensitive to alignment owing to the retro-reflective nature of the beads in the RRS.

Ideally the illuminating LED would be placed on axis along with the camera optic, however this is practically difficult owing to the high NA needed in both illumination and imaging at approximately 0.5. The LEDs are therefore positioned outside the camera optic's aperture leading to pairs of images from individual mosquitoes. A direct image of the mosquito is seen by the camera against the bright RRS and a shadow image is formed from the slightly off-axis LED on the RRS. This effect enables triangulation and hence recovery of 3D mosquito position data [26] but requires complex calibration and extensive signal processing. In contrast, it has been demonstrated that 2D tracking provides entomologically useful information [1, 2, 4] and the requirement in malaria control is to rapidly test design iterations of interventions over long time periods (hours) and with sufficient repeats to give statistically reliable data. Hence, this paper targets robust 2D tracking.

For 2D tracking over the 2 m depth of field needed here only the direct image is required. The contrast of the direct image is higher than that of the shadow images due to the extended distance over which diffraction occurs for the latter. Furthermore, each LED within the ring light will form shadows in different spatial locations of the image, whereas the direct image is in the

same location and becomes reinforced with each additional LED. A custom ring light source was constructed with 12 OSRAMTM SFH 4235 infrared LEDs (peak wavelength 850 nm) [27]. The wavelength provides good sensitivity for monochrome silicon based detectors and there is no evidence from previous back-lit experiments or on-going studies with the RRS setup that the direction of the illumination and orientation of the human bait affected mosquito behaviour [1, 6].

The inset in Figure 1b shows a photograph of the ring light and mounting arrangement via an optical rail that allows independent adjustment of LED plane with respect to the camera and lens. RRS based imaging experiments were conducted with 12 MPixel cameras, either Ximea CB120RG-CM (used with Canon EF 14mm f/2.8L II USM lenses) or Dalsa FA-80-12M1H (used with Nikon 14mm f/2.8D AF ED Lens), operating at 50 fps and with 14 mm focal length camera lenses. The optical system (camera lens, Fresnel lens and RRS screen) is approximately telecentric, with the deviations coming from the steps in the Fresnel lens (which makes such a large aperture lens physically manageable) and the RRS which gives a cone of reflected light, partly compensated by closing the camera lens aperture to typically F8.0. The optical setup provides approximately 0.5 mm per pixel which gives adequate sampling of the *Anopheles gambiae* mosquito which is typically 2.5-3.5 mm wing length (other species can be much larger). Hence, in the images there is a bright background with the mosquitoes appearing as dark images. In the RRS approach there is a double pass of the measurement volume giving increased diffraction which limits the practical depth of field to approximately 2 m.

(b) Signal Processing

In order to interpret mosquito behaviours it is imperative to be able to track them when swooping above or around the bednet as well as when host seeking and probing the net. To form contiguous tracks across all the different areas, a robust segmentation process is needed. When the mosquitoes are in the region above the bednet there is at least 3-5 greyscales difference between the mosquito image and the background. The contrast reduces markedly when the mosquito is either in front or behind the bednet. With the RRS setup the light passes each layer of the bednet twice (see Figure 1b) leading to 2-3 greyscales difference between the mosquito and background when the mosquito is in front of the net (closer to the Fresnel lens) and about 1-2 greyscales for mosquitoes behind the net. Graphs of the typical intensity distribution around a mosquito image are given in Figure 2 in both 3D (inverted intensity scale) and image formats. Figure 2a shows a typical mosquito shadow image using back-lit imaging when the mosquito is outside the bednet, the corresponding image with RRS imaging is in Figure 2b. Significantly reduced contrast can be seen in Figure 2c which is a back-lit mosquito image from across the bednet and Figure 2d from the RRS approach. Mosquito detection relies on the contrast between the bright background and dark image of the foreground (mosquitoes). Objects that do not allow light transmission (e.g. the human host, the bed, etc.) prevent mosquito detection in front or behind such objects similarly for both backlit and RRS setups.

The original approach to segmentation [5] used a difference image between consecutive frames and a single threshold value to identify movement. Whilst developing the RRS system, two issues were found. Firstly, the contrast of a mosquito image in low intensity areas is similar to the camera noise. Secondly, in high intensity regions, the noise level and contrast is higher necessitating a different threshold. Noise reduction is necessary to enable robust thresholding of mosquito images in low intensity areas. Several filter types were considered (Gaussian, median etc) and applied to original or difference images. Using noise reduction effectiveness and computational overhead as criteria, a Gaussian filter applied to the difference image was selected using a 15 x 15 pixel kernel (implemented via the OpenCV library [28]). The width of the Gaussian function is defined by the standard deviation as [28]:

$$\sigma = 0.3 \cdot ((ksize - 1) \cdot 0.5 - 1) + 0.8.$$

(a) Backlit recording setup

(b) Retroreflective recording setup with an inset photograph of the ring light mounted together with the camera on a standard photographic tripod.

Figure 1: Schematic diagram of the original backlit and new retroreflective screen recording systems.

The filter values were scaled such that the integral across the whole filter was 1. The typical benefit of the filter can be observed in Figure 3 which shows as red the pixels above a user defined threshold before and after application of the filter, in this example the threshold was 2 greyscales.

Figure 2: Mosquito intensity distributions under back-lit or RRS illumination and for mosquito positions above the bednet or in front of the bednet. 3D maps use inverted intensity scale.

Figure 3: The typical effect of de-noising on part of a difference image containing a mosquito image. Both cases show the same part of the frame. The red colour shows differences greater than 2 greyscales.

To address the need for variations in threshold to segment a mosquito image, the high and low intensity areas can be classified from examination of a normalised histogram. For the RRS system, the greyscale ranges associated with the different regions of the image can be seen in Figure 4b. The two regions of interest for mosquito flight are the background which is above or to the side of the bednet and the bednet region itself, either in front or behind the bednet. A manual definition

of the threshold that separates these regions is required to run segmentation algorithm and is typically around 40-60 greyscales and is consistent for both backlit and RRS imaging approaches.

The system is insensitive to the ambient lights level (if it does not create direct reflection on the Fresnel lens). Thresholding is applied to the difference image obtained between nearby frames in the temporal sequence, hence the threshold level is largely insensitive to variations in ambient light. The image obtained is dominated by the IR illumination directed to the RRS and hence reflected into the camera. Furthermore, field experiments are usually done at night, with closed windows and doors and hence the threshold required for successful segmentation is stable. So, ambient light level encountered in typical experiments cannot affect the selected thresholds.

It can be seen from Figure 4 that the RRS system gives larger regions of high intensity in the background whereas this region is more widely distributed for the backlit case. The bednet region has a larger greyscale range for the backlit system (showing less uniform illumination in the backlit case), but its peak is at a higher greyscale, approximately 40, than for the RRS system. It can also be seen that the darkest regions, corresponding to the human bait are brighter for the RRS by circa 10 greyscales than in the backlit case. This occurs due to the direct front illumination which will also be incident on the mosquitoes and can slightly reduce the contrast in comparison with the background.

(a) Back-lit image histogram

(b) RRS image histogram

Figure 4: Normalised image histograms with different illumination approaches.

The updated segmentation algorithm incorporates the following stages.

- (i) Form a difference image between the n th and $(n-i)$ th images. Typically i is set to 5 frames to aid discrimination. This produces a pair of detected mosquito images with opposite signs and their separation is the mosquito displacement between the two frames.
- (ii) A large kernel, typically 9×9 to 21×21 pixel, Gaussian filter is then (optionally) applied to the difference image to reduce noise.
- (iii) For the brighter, background region, the algorithm searches for all regions of positive difference. An image threshold is calculated and applied to segment the mosquito image. Morphological operators can be optionally applied to improve the outer contour of the mosquito image (opening and closing). Statistics are then calculated: the centre coordinates, the area, i.e. standard 'blob' analysis. Dilation performed by 5-7 pixels aid the detection of broken down images. Mosquito images are removed with either too large or small area (based on user defined thresholds). The result is a set of candidate mosquito images that satisfy the selection criteria and their positions in the image. The typical threshold used for the bright areas is 3 greyscales.
- (iv) The process is repeated for the darker image regions containing the bednet. The threshold used to segment the mosquito images tends to be lower at 1 to 2 greyscales.
- (v) The data from the two passes (the brighter and darker image regions) are then combined together.

Apart from the addition of a Gaussian filter for de-noising and classification of the region to determine the appropriate threshold, the mosquito segmentation algorithm is identical to the one described in [5]. Whilst more complex algorithms could be implemented, the requirement to control the computational overhead was important given the need for operation in the field and on data sets of approximately 2 TB per hour of recording from the two cameras.

3. Results and discussion

(a) Evaluation of Illumination quality

Exemplar images from the backlit and RRS illumination systems are given in Figure 5. The ring structure from the pair of Fresnel lenses used in the backlit approach are visible across the image in Figure 5a, whereas this structure is averaged out when a single Fresnel lens is used with the RRS in Figure 5b. The horizontal banding in the RRS image derives from the 50 mm wide adhesive tape obtained from 3M and the reflections from the LED ring light give small localised areas where individual mosquito images cannot be segmented. In the backlit case the roof of the bednet is tilted whereas with RRS the data were obtained using a flat bednet roof. Each figure shows 3 intensity profile graphs in the region above the bednet, i.e. the background. It can be observed that with the RRS the intensity towards the edge and corners of the image is maintained at a higher level, whereas with backlit imaging the intensity drops to <50% of the central area. It is clear that in the corners of the frame from the backlit system the illumination level is very similar to that in the middle of the bednet (see Figure 6a). This will cause reduced mosquito detection in the corners of the frame and difficulty in defining a robust threshold.

The ring structure from the two Fresnel lenses can also be seen in the profiles from Figure 5a with backlit imaging and for unstable field recordings can lead to erroneous artefacts in the difference images used for mosquito segmentation.

The temporal stability of the illumination using the bespoke LED ring light source was compared with that of the original single LED used in the original backlit system. Statistical analysis over 250 frames (5 seconds) of data in different parts of the image showed the expected result that the standard deviation and greyscale range increase with mean intensity. The cameras used for backlit imaging and those in the RRS experiments both using the same type of CMOS

(a) Image from backlit system

(b) Image from retroreflective screen system

Figure 5: Spatial distribution of light levels across the background of the image along three different directions using line probes. Line probes are parameterised by a coordinate from 0 to 1, where 0 is at the centre of the image and 1 is at the edge.

(a) Image from backlit system

(b) Image from retroreflective screen system

Figure 6: Spatial distribution of light levels across the bednet along three different directions using line probes. Line probes are parameterised by a coordinate from 0 to 1, where 0 is at the centre of the image and 1 is at the edge.

detectors and gave average levels of 1.84% greyscale standard deviation as a proportion of the mean intensity.

(b) Quantitative performance metrics

The performance of the combined imaging and segmentation algorithms has been quantified using a number of metrics, thus enabling a quantitative comparison of the backlit and RRS imaging setups. The data are presented in Table 1. Each dataset was obtained from a typical 1 hour recording with the cameras operating at 50 frames per second.

The recording made with backlit system was done using Tiassale strain (an insecticide resistant variety) of *Anopheles gambiae* mosquitoes. There were 25 female mosquitoes, between 3 and 5 day old, unfed and deprived of sugar for 4 hours prior to filming and left to acclimatise in the filming room for 1 hour before testing. The test was conducted within 1-3 hours of the start of scotophase (i.e. their night time). The male human host was clothed, barefoot and lay on its back and as immobile as comfortably possible. This test was done at 28 degree C and 78% relative humidity (RH).

The recording with RRS system was done using N'guosso strain (an insecticide susceptible variety) of *Anopheles Coluzzicoluzzii* mosquitoes. There were also 25 mosquitoes, between 3 and 7 days old, deprived of sugar 1 day before and of water 5 hours before testing. Mosquitoes were placed in the room 1 hour before testing to acclimatise. Mosquitoes were kept and the test was done in the same climate, i.e. at 27 ± 2 degrees C and $70 \pm 10\%$ RH under a 12hr light/12hr dark cycle.

Both tests used modified PermaNet 2.0 LLIN bednets (55 mg/m^2 deltamethrin; Vestergaard, Lausanne, Switzerland). The modification involved making the roof tilted, so, that the roof will be visible in the view of the camera. In both cases all mosquitoes were released at the beginning of the test from a cup located 2 metres above the floor and about 1.5 metres from the edge of the bednet.

The same recording (made with RRS recording system) was used for the Figure 8.

It is important to emphasise that the same algorithms have been used to process the recordings from both the backlit and RRS systems (with small variations in parameters to optimise the results). Therefore, differences in performance are due to the optical setup. Notably, the greyscale threshold for detection of mosquito images across the challenging bednet region was the same (1 greyscale). An initial tracking process was performed [5] allowing only a single missing mosquito position within a track and discarding any tracks with less than 5 positions; hence the number of 'detected positions in tracks' in Table 1 is determined. A second tracking iteration enabled existing tracks to be lengthened and tracks to be combined together with gaps of up to 15 frames between consecutive positions in a track.

Any gaps in a track, i.e. where the time between consecutive track positions is a multiple of the inter-frame time, Δt , indicate a missed mosquito position(s). Some missed positions occur as mosquitoes fly across completely occluded areas but others are due to locally poor contrast. The total number of potential positions in a track, N_t , is calculated from:

$$N_t = \sum_{k=0}^{n-1} \text{Int} \left[\frac{\Delta t_k}{\Delta t} \right],$$

where n - number of recorded positions, Δt_k - time difference between the current and previous recorded position, $\text{Int}[\dots]$ gives the nearest integer value. An individual gap in a track is identified when $\text{Int} \left[\frac{\Delta t_k}{\Delta t} - 1 \right] > 0$ and hence the total length of gaps in a track, N_{Gaps} , is given by:

$$N_{Gaps} = \sum_{k=0}^{n-1} \text{Int} \left[\frac{\Delta t_k}{\Delta t} - 1 \right],$$

The data in Table 1 provides average and standard deviation of gap size as well as the number of gaps as a proportion of the number of detected positions (to normalise the expected experimental variability). These metrics are determined for the background (outside the bednet) and the across the bednet regions. Statistics are also given for the average track length and average track length without gaps. The latter is the average length of track pieces without any gaps, i.e. where there is a detected position at every time step. Finally, mosquito detection percentages are given from the ratio between number of detected positions used in tracks and total number of potential positions, N_t .

(c) Discussion

The rationale for the RRS based imaging system has been described above and it is clear that this approach delivers a smaller operational footprint as well as only needing optical access from one side of the measurement volume. It is worth noting that the backlit system has a very small optical power density where the scene volume was 4.8 m^3 and total light source power output is 0.88 W (for two LEDs), thus, light power density is 0.18 W/m^3 . The new system provides

¹Retroreflective screen system

Table 1: Performance metrics of mosquito position detection for backlit and retroreflective screen systems.

Metrics	Systems	Backlit [5]	RRS ¹
Detected positions in Tracks		19128	22483
Mean \pm SD Gap Size (Frames) Across Net		6.88 \pm 11.91	4.19 \pm 4.22
Gaps/Positions Ratio Across Net		0.066	0.018
Mean \pm SD Gap Size (Frames) Outside Net		4.18 \pm 6.58	4.58 \pm 4.62
Gaps/Positions Ratio Outside Net		0.036	0.017
Average Track Length		78.02	77.54
Average Uninterrupted Track Length		16.82	29.85
Detected Mosquitoes Across Net		54%	92%
Detected Mosquitoes Outside Net		85%	92%
Detected Mosquitoes Overall		69%	92%

slightly larger scene volume (due to greater height of the Fresnel lenses) of 5.9 m³, but light source power output is approx. 12 W (for two LED ring lights) and, thus, the optical power density is 2 W/m³. Repeat experiments and rotation of the human bait show that the increased power density from the fixed position of the illumination does not influence the behaviour of the mosquitoes. There are additional benefits in uniformity of the illumination into the corners of the measurement volume compared to the backlit setup as evidenced by the spatial distribution data in Figures 5 and 6.

In contrast, backlit illumination causes a hot spot in the middle of the image and any increase in source power will cause saturation and prevent mosquito detection. There is the opportunity with the more uniform illumination from RRS imaging on the other hand, allows to the light level to be raised significantly further without causing saturation of the image in the centre.

The 12 LEDs in the ring light are necessarily displaced from the camera lens axis and therefore create collimated beams that traverse the measurement volume at a small angle, $<4^\circ$, to the optical axis of the Fresnel lens. This reduces any shadowing caused by other objects, such as the bednet, affecting the image of the mosquitoes. The position and size of the diffuser in the RRS system (being the full aperture RRS itself) gives increased scattering compared to that in the backlit setup. The scatter from the RRS having a larger angular subtense than the propagation directions of the illumination gives a diffuse cone of illumination of $>10^\circ$. These effects are probably key to giving increased mosquito segmentation performance over the critical bednet areas, leading to the level of mosquito detection across the bednet for RRS of 92% compared to 54% for backlit imaging (Table 1). The practical benefits are important for ease of use in field settings, e.g. in Banfora, Burkina Faso. The RRS approach requires 2 Fresnel lenses (rather than 4 for the backlit case) and two tripods for the camera-LED assemblies (4 tripods for two cameras and two light sources for backlit). The RRS itself can be assembled in the field from locally sourced plywood sheets and the retro-reflective tape bonded to it.

The performance of the RRS approach to track mosquitoes in front of and behind the bednet has been evaluated in lab experiments where the mosquitoes are released via an aspirator at the specific locations. Segmented and tracked results are given in Figure 7a when the mosquitoes are released in front of the bednet and in Figure 7b where the point of release is behind the bednet. The image of the aspirator at the moment of release is clearly visible on the right edge of each image. Several mosquitoes were released simultaneously in each case. It is clear that mosquitoes located closer to the Fresnel lens (in front of the net) are reliably identified by the segmentation algorithm everywhere – the tracks show a regular spacing of segmented positions. Mosquitoes flying closer to the retroreflective screen (behind the net) are consistently identified above the net

and across the vertical net walls. Segmentation performance is slightly worse across the inclined net roof, although most positions are still identified and sufficient to enable flight path tracking. This occurs due to the combination of limited depth of field and additional optical attenuation through the inclined net which gives increased occlusion. Also, at the beginning of the tracks mosquitoes were not identified well because they were blown from the aspirator quite fast (much faster than their natural maximum velocity).

(a) Positions and tracks of mosquitoes flying in front of the bednet. (b) Positions and tracks of mosquitoes flying behind the bednet.

Figure 7: Segmented positions of the mosquitoes and flight tracks released using an aspirator (visible on the right of each frame) specifically in front (a) and behind (b) the bednet.

Figure 8 shows identified mosquito positions from both cameras with a human volunteer lying under the net. Qualitative inspection reveals a high level of activity in front and behind the bednet as well as bouncing behaviours particularly evident above the feet of the person (left hand side but with some continuity to the right hand camera view). These data were obtained with Ximea CB120RG-CM cameras. The left hand camera used 5 ms exposure time with no gain, whereas the right hand camera was used with 5 ms exposure and -3.5 dB gain. The continuity of tracks between the two views reflects the robustness of the segmentation algorithm to these varying conditions. Movement of the volunteer and draughts cause movement of the bednet and the associated strings supporting it and lead to some false segmentation in individual frames that are largely filtered out via the tracking algorithms. Supplementary video file demonstrates short part of the recording, where mosquitoes move from one side to the other and some bouncing activity on the roof of the net.

The quantitative performance metrics presented in Table 1 are from different experiments for the backlit and RRS imaging systems and hence exhibit natural variability due to the use of different mosquitoes and human bait. This is evident in the detected positions in tracks metric which shows that the RRS experiment produced a higher number of segmented mosquito positions that passed the consistency requirements to be part of mosquito flight tracks. The data

Figure 8: Example of full RRS system segmented recordings using Ximea CB120RG-CM cameras and 5 ms exposure per frame. Left camera, no gain, right camera, -3.5 dB gain.

on gap size in tracks shows that the RRS imaging approach gave similar performance both across net and outside net regions. In contrast the backlit imaging setup gives larger track gaps across the bednet probably due to reduced illumination levels away from the centre of each camera's field of view and the reduced transmission of the bednet in this region. The variability in gap size (given by the standard deviation, SD) is higher for both regions with backlit imaging, but especially so for the across net region. The data for the number of track gaps normalised by the number of detected positions shows similar trends: consistent performance from the RRS setup in both across net and outside net regions and worse metrics with backlit imaging.

The track length data shows that average performance of backlit and RRS imaging systems is similar. However, the uninterrupted track length is significantly greater for RRS than backlit, indicating the higher consistency in segmenting mosquito positions. The mosquito detection percentages show that the RRS imaging approach achieves consistent performance in both across net and outside net regions with detection levels >90% that are significantly higher than for backlit imaging. It is worth noting again that the same processing pipeline was applied to the recordings made with backlit and RRS recording systems. So, the differences are due to the optical setup rather than signal processing. With this improved performance in mosquito image segmentation, the tracking algorithms are able to reliably bridge the gaps in tracks that do occur and hence deliver significantly improved flight tracks for entomologists.

The global threat to public health from mosquito-borne diseases like malaria, dengue, yellow fever and emerging arboviral infections such as Zika, together with the spread of insecticide resistance, demonstrate the unabated challenge for mosquito control and the relentless need for novel control tools. Tracking has already proven its value for advancing bednet design [4], but the performance of the RRS imaging approach has broad potential for numerous studies, ranging from basic research to evaluation of novel control tools. A widely applicable example is the tracking of mosquitoes or other flying insects during approach and landing on single or multiple test targets. Binary choice-tests are used widely to measure the effects of repellents or other

attractants on mosquitoes, or genetic or phenotypic differences between mosquito populations or laboratory strains. Figure 9 illustrates the potential of the RRS system to generate images suitable for a typical study comparing attractiveness to mosquitoes of two human subjects, e.g. to quantify the degree to which a repellent treatment on one host kills or possibly diverts mosquitoes to the untreated hosts. is being used in both field and lab environments to further understand the interaction between bednets and mosquitoes, including the testing of novel designs [4]. In addition, the system is helping the definition of new protocols for commercial bednet assessment. The RRS system has also been applied to a new study to assess the variation on attractiveness of individuals to mosquitoes. The experimental design is to have two people within the field of view but seated separately. Host seeking mosquitoes respond to the person giving off higher levels of attractant and then blood feed. An initial image showing segmented mosquitoes is given in Figure 9

Finally, the more compact and easily aligned RRS system greatly facilitates transportation between and installation at multiple sites, including rural or relatively remote sites. Multiple RRS tracking systems are now being installed in east and west Africa where they will be used to characterise the efficacy and monitor the progress of the so-called ‘next generation’ of insecticidal bednets; essential evidence for decision-makers planning effective malaria control of pyrethroid-resistant mosquitoes. In this example, the individual on the left hand side is clearly more attractive to mosquitoes than that on the right, with several mosquitoes exhibiting bouncing type behaviour in identifying a location to blood feed. This type of experiment could be extended to evaluate the performance of repellents and other interventions:

Figure 9: Exemplar image captured using the RRS recording system described herein, showing 10 minutes flight activity by 10 *Anopheles gambiae s.l.* females around two seated adults. Higher flight activity is apparent at the person on the left, who was bitten twice while the individual on the right received three bites..

4. Conclusions

The introduction of a retro-reflective screen to large volume, human baited mosquito tracking studies delivers significant improvements to mosquito detection rates in comparison to previous backlit imaging approaches. This is primarily due to increased uniformity of illumination and the narrow angle diffuse nature of the reflected light from the RRS. A disadvantage of RRS imaging is that the light double passes each bednet layer that is typically (although, not necessarily) the focus of the experiments, reducing the available light intensity. However, the use of large kernel Gaussian noise reduction filters combined with low detection thresholds of 1 to 2 greyscales have been shown to give robust mosquito detection. With RRS imaging, mosquito detection rates of 92% have been obtained, compared to an average of 69% with backlit imaging.

There are further practical benefits of the RRS imaging setup in terms of reduced footprint, reduced number of optical components required and reduced sensitivity to mis-alignment due to the retro-reflective nature of the material.

RRS based imaging setups are a powerful tool in entomological research, especially for species like the *Anopheles* mosquitoes that transmit malaria, nocturnally active mosquitoes that require remote detection as their natural behaviour is influenced by the presence of a human observer. RRS systems are installed in research laboratories in the UK and Burkina Faso, where the insight they provided on malaria vector behaviour led directly to novel next-generation bednet designs and an innovative solution to the threat from insecticide resistance [4].

Acknowledgements

The authors would like to thank the volunteer sleepers both in the laboratory in Liverpool and in the field in Burkina Faso and are especially grateful to the community at Tengrela, Burkina Faso, who continue to support studies at their village. Also, the authors are very grateful for all constructive comments from the reviewers, which greatly helped to improve the paper.

Ethics

All research methods were performed in accordance with approved guidelines for those procedures and written informed consent was obtained from all volunteer subjects. Video recording work was approved by the Research Ethics Committees at the Liverpool School of Tropical Medicine (LSTM Research Protocol 16-38, 11 October 2016, Liverpool) and Centre National de Recherche et de Formation sur le Paludisme (CNRFP Deliberation no. 2016-9-097, 20 September 2016, Ouagadougou). No adverse effects of treatment or mosquito-borne infections were reported by volunteers during the course of the study.

Data, code and material

Image segmentation software source code is available at [29], All other custom codes (~~not already published elsewhere~~) and all data used in this work are available at [30] and larger datasets used for illumination analysis are available at [31].

Competing interests

The authors declare no competing interests.

Funding

This research was supported with funding from the UK Medical Research Council (MRC) and the UK Department for International Development (DFID) under the MRC/DFID Concordat

agreement (MR/M011941/1), a Wellcome Trust Collaborative Award (Malaria in Insecticide Resistant Africa MiRA; 200222/Z/15/Z) and the Bill & Melinda Gates Foundation (OPP1159078).

Authors contributions

C.K., D.P.T., C.E.T. and V.V. conceived the idea of the improved system. V.V., C.K. and C.S. developed the new system and did initial testing. P.J.M. and G.M. gave advice on specifics of entomological use and field conditions for the system. G.M. and V.V. setup both field and lab based systems and did initial testing of the system at the field site. G.M. collected field test data, whereas A.G. collected test data from lab based system. C.K., C.S., and V.V. did initial pre-processing and initial assessment of the collected data. V.V. together with C.K., C.E.T., and D.P.T. developed and implemented assessment metrics and methods. C.K. with the help of C.S., C.E.T., D.P.T., and V.V. devised and implemented improvements to the segmentation algorithm. V.V. with the help of C.K. and C.S. ran the performance analysis. V.V., C.K., C.E.T and D.P.T. wrote the paper with contributions from P.J.M. All authors approved the final submitted version.

References

- 1 Parker JEA, Angarita-Jaimes N, Abe M, Towers CE, Towers D, McCall PJ. Infrared video tracking of *Anopheles gambiae* at insecticide-treated bed nets reveals rapid decisive impact after brief localised net contact. *Sci Rep.* 2015;5. Available from: <https://doi.org/10.1038/srep13392>.
- 2 Parker JEA, Angarita Jaimes NC, Gleave K, Mashauri F, Abe M, Martine J, et al. Host-seeking activity of a Tanzanian population of *Anopheles arabiensis* at an insecticide treated bed net. *Malar J.* 2017;16. Available from: <https://doi.org/10.1186/s12936-017-1909-6>.
- 3 malERA Consultative Group on Vector Control T. A Research Agenda for Malaria Eradication: Vector Control. *PLOS Medicine.* 2011 01;8(1):1–8. Available from: <https://doi.org/10.1371/journal.pmed.1000401>.
- 4 Murray G, Lissenden N, Jones J, Voloshin V, Toé H, Sherrard-Smith E, et al. Novel bednet design targets malaria vectors and expands the range of usable insecticides. *Nat Microbiol.* 2020;5:40–47. Available from: <https://doi.org/10.1038/s41564-019-0607-2>.
- 5 Angarita-Jaimes NC, Parker JEA, Abe M, Mashauri F, Martine J, Towers CE. A novel video-tracking system to quantify the behaviour of nocturnal mosquitoes attacking human hosts in the field. *J R Soc Interface.* 2016;13. Available from: <https://doi.org/10.1098/rsif.2015.0974>.
- 6 Gibson G. A behavioural test of the sensitivity of a nocturnal mosquito, *Anopheles gambiae*, to dim white, red and infra-red light. *Physiol Entomol.* 1995;20. Available from: <https://doi.org/10.1111/j.1365-3032.1995.tb00005.x>.
- 7 Spitzen J, Takken W. Keeping track of mosquitoes: a review of tools to track, record and analyse mosquito flight. *Parasites & Vectors.* 2018 Mar;11(1):123. Available from: <https://doi.org/10.1186/s13071-018-2735-6>.
- 8 Bhatt S, Weiss D, Cameron E, Bisanzio D, Mappin B, Dalrymple U, et al. The effect of malaria control on *Plasmodium falciparum* in Africa between 2000 and 2015. *Nature.* 2015;526(7572):207–211.
- 9 World Health Organisation: World Malaria Report, WHO, Geneva; 2019. Available from: <https://www.who.int/publications-detail/world-malaria-report-2019>.
- 10 Back Malaria Partnership. Global Malaria Action Plan, WHO, Geneva; 2008. Available from: <http://archiverbm.rollbackmalaria.org/gmap/gmap.pdf>.
- 11 Huho B, Briët O, Seyoum A, Sikaala C, Bayoh N, Gimnig J, et al. Consistently high estimates for the proportion of human exposure to malaria vector populations occurring indoors in rural Africa. *International Journal of Epidemiology.* 2013 02;42(1):235–247. Available from: <https://doi.org/10.1093/ije/dys214>.
- 12 Bayoh MN, Walker ED, Kosgei J, Ombok M, Olang GB, Githeko AK, et al. Persistently high estimates of late night, indoor exposure to malaria vectors despite high coverage of insecticide

- treated nets. *Parasites & Vectors*. 2014 Aug;7(1):380. Available from: <https://doi.org/10.1186/1756-3305-7-380>.
- 13 Long lasting mosquito net PermaNet 2.0, Vestergaard Frandsen;. Available from: <https://www.engineeringforchange.org/solutions/product/permanet-2-0/>.
 - 14 Durable long-lasting mosquito net with fast action against mosquitoes and additional efficacy against Pyrethroid-resistant mosquitoes Olyset[®] Plus, Sumitomo Chemical Co. Ltd.; Available from: <https://sumivector.com/sites/default/files/Olyset%20Plus%20-%20Technical%20Brochure%202018%20-%20English.pdf>.
 - 15 Manoukis NC, Butail S, Diallo M, Ribeiro JMC, Paley DA. Stereoscopic video analysis of *Anopheles gambiae* behavior in the field: Challenges and opportunities. *Acta Tropica*. 2014;132:S80 – S85. Available from: <http://www.sciencedirect.com/science/article/pii/S0001706X13001769>.
 - 16 Hawkes F, Gibson G. Seeing is believing: the nocturnal malarial mosquito *Anopheles coluzzii* responds to visual host-cues when odour indicates a host is nearby. *Parasit Vectors*. 2016;9. Available from: <https://doi.org/10.1186/s13071-016-1609-z>.
 - 17 Butail S, Manoukis N, Diallo M, Yaro AS, Dao A, Traoré SF, et al. 3D Tracking of Mating Events in Wild Swarms of the Malaria Mosquito *Anopheles gambiae*. *Conf Proc IEEE Eng Med Biol Soc*. 2011;.
 - 18 Butail S, Manoukis N, Diallo M, Ribeiro JM, Lehmann T, Paley DA. Reconstructing the flight kinematics of swarming and mating in wild mosquitoes. *J R Soc Interface*. 2012;9. Available from: <https://doi.org/10.1098/rsif.2012.0150>.
 - 19 Butail S, Manoukis NC, Diallo M, Ribeiro JMC, Paley DA. The dance of male *Anopheles gambiae* in wild mating swarms. *J Med Entomol*. 2013;50. Available from: <https://doi.org/10.1603/ME12251>.
 - 20 Mendikute A, Yagüe-Fabra JA, Zatarain M, Bertelsen Á, Leizea I. Self-calibrated in-process photogrammetry for large raw part measurement and alignment before machining. *Sensors*. 2017;17(9):2066.
 - 21 Towers CE, Towers DP, Jones JD. Optimum frequency selection in multifrequency interferometry. *Optics letters*. 2003;28(11):887–889.
 - 22 Straw AD, Branson K, Neumann TR, Dickinson MH. Multi-camera real-time three-dimensional tracking of multiple flying animals. *J R Soc Interface*. 2011;8(56):395–409. Available from: <https://doi.org/10.1098/rsif.2010.0230>.
 - 23 McCall PJ. Mosquito bed net assembly; 2014. Patent number EP3062663B1. Available from: <https://patents.google.com/patent/EP3062663B1/en>.
 - 24 NTKJ CO., LTD. (Japan) CF1200 Fresnel lens;. Available from: <https://www.ntkj-japan.com/fresnel-lens/>.
 - 25 3M[™] ScotchLite[™] High Gain Reflective Sheeting; 2015. Available from: <http://multimedia.3m.com/mws/media/10472160/3m-scotchlite-high-grain-reflective-sheeting-7610.pdf>.
 - 26 Kröner C, Towers CE, Angarita-Jaimes N, Parker JEA, McCall P, Towers DP. 3D tracking of mosquitoes: A field compatible technique to understand malaria vector behaviour; 2016. Heidelberg, Germany.
 - 27 OSRAM[™] Platinum DRAGON[®], SFH 4235; 2018. Available from: https://www.osram.com/ecat/Platinum%20DRAGON%20%ae%20SFH%204235/com/en/class_pim_web_catalog_103489/global/prd_pim_device_2219732/.
 - 28 Bradski G. The OpenCV Library. *Dr Dobb's Journal of Software Tools*. 2000;.
 - 29 Github repository with image segmentation custom code;. Available from: <https://github.com/kroener/SeqFileProcessing2D>.
 - 30 Github repository with illumination and performance analysis code and data;. Available from: <https://github.com/Pigrenok/TrackingPerformance>.
 - 31 Two StreamPix Sequence v.5 files, which contains dataset used for illumination analysis;. Doi: [10.6084/m9.figshare.8534057](https://doi.org/10.6084/m9.figshare.8534057).

Response to referees' comments

Comment from referee 1

Reducing transmission of mosquito-borne illness requires simple and robust methods to quantify behavior in natural settings. An example is measuring flight trajectories around a bednet covering an individual, but current tracking methods suffer from non-uniform illumination, which complicates automated tracking due to changing background light levels. Here the authors introduce a method for providing much more uniform illumination via a retro-reflective lens behind the bednet. This substantially improves tracking, in particular enabling longer continuous detection of flight trajectories, which could benefit quantitative behavioral analysis.

This work provides a useful methodological advance in collecting insect flight data and could improve robustness of data collection and analysis in conditions outside the laboratory. The technical language is precise, which will enable others to easily reproduce the setup.

The major addition that would improve the scientific value of the paper would be a demonstration of the method to yield a novel scientific finding. One idea would be to expand on the "attractiveness" question exemplified in Figure 9. What makes one person more attractive to mosquitoes than another is important to understand, and it would be interesting to see quantitatively how the retro-reflective screen method improves one's ability to assess differences in attractiveness. It would also be interesting to use this method, along with more detailed behavioral analysis, to quantify distributions of mosquito behaviors and see how they changed in different conditions. E.g. which behaviors do mosquitoes engage in more frequently when in the presence of an attractive host? Or which behavioral patterns differentiated disease-carrying vs non-disease-carrying mosquitoes?

The referee refers to Figure 9 of the manuscript and is correct in highlighting the importance of understanding an individual's attractiveness to mosquitoes. The authors felt it was worth publishing the detail on the tracking system separately from the scientific findings on mosquito behaviour and devices to mitigate disease transmission. The audiences for these two sets of information have different expertise and there is significant detail needed in each case. Hence, the most recent entomology findings are reported in reference [4] of the current manuscript in which the efficacy of the 'barrier bednet' against both insecticide resistant and insecticide susceptible mosquitoes is given; built on data obtained with the RRS imaging system as well as the original backlit approach. The data in Figure 9 was added following feedback from one of the referees to our original Royal Society Interface Journal submission – the referee commenting that application to human baited bednets was very niche – hence our desire to show more general application to other questions in entomology. Within the text of our Open Science submission we had attempted to make it clear (content on pages 4 and 14) that Figure 9 was an example from initial experiments to look at individual attractiveness.

The increased segmentation and hence tracking performance of the RRS approach compared to the original backlit method yields higher quality tracks – as has been quantified in detail in this submission for the human baited bednet data. In using the RRS system to assess individual attractiveness it is the equally important benefits of ease of use, simple alignment and high quality data that made the experiments straightforward to achieve and would not have been possible with the backlit method. The referee is correct to realise that individual attractiveness will depend on several factors; we have added some detail regarding the experiment that led to the tracks in Figure 9. In this test 10 female *Anopheles gambiae* were released and tracks were recorded for 10 min at two seated adults. Track colour indicates time active (blue=early; red=late). Despite higher activity, the left host received 2 bites, the right host received 3. We have also added a more thorough discussion of the factors that are thought to affect individual attractiveness within the discussion, however, the referee is correct that a wide ranging experimental programme would be needed coupled with new behavioural analysis in order to draw conclusions and this is a subject for future research.

Minor notes/questions:

1. The technical writing was precise but a bit verbose. E.g. “Large field of view back lit imaging systems have been reported with two parallel imaging channels to give a measurement volume of $2 \times 2 \times 1.4$ m in total with large aperture Fresnel lenses enabling collimated illumination and telecentric imaging” is a difficult first sentence of a paragraph to parse. It could improve the manuscript to put more emphasis up front on the core advance of using a retro-reflective screen to create more uniformly distributed light and allocate details to the methods section or an appendix.

Thank you for noting this long and complex to comprehend sentence. The sentence was broken down into several shorter sentences which are now easier to read. In general, this particular paragraph forms part of the introduction and is dedicated to the previous backlit system, describing why it is unique among other tracking systems and what drawbacks it has; hence forming the motivation to the development of the new RRS tracking system.

2. On P2 L35, it's stated that mosquito images are occluded when the mosquito is in front of the bednet. I don't understand why occlusion would occur if the bednet is behind the mosquito.

The term "occlusion" here is used in a wider context, referring to the mosquito visibility (contrast) in the image. The light is reflected from the RRS and is occluded by the bednet before arriving at the mosquito and hence will reduce the contrast of the mosquito image compared to the background. The text has been updated on page 2 accordingly to clarify the phenomenon.

3. On P3 L24, the phrase “per camera” is used, suggesting there are multiple cameras in this setup. However, it was my understanding that there is only one camera involved when the retro-reflective screen method is used.

An explicit statement has been added on page 3 to explain that two cameras with non-overlapping images are used both in backlit and in RRS systems. Each camera records the image field from one Fresnel lens, which covers an area of about 1.4×1 m. To cover a whole bednet, two Fresnel lenses standing side by side are needed to give a total field of view of 2×1.4 m, and thus two cameras are needed. The system is scalable to handle larger areas either by using larger Fresnel lenses or by adding extra cameras.

4. Would the illumination level used in this method be expected to affect mosquito behavior at all? Or is it far enough in the IR spectrum as not interfere?

As far as the authors are aware, mosquitoes are not affected by IR illumination at around 850 nm wavelength. The statement highlighted in olive colour on page 5 addresses exactly this concern. This is consistent with the earlier work by Gibson in 1995 [6]. Repeated experiments with the RRS setup have been conducted over a number of years in which the orientation of the human bait has been varied in order to verify that mosquito behaviour patterns vary accordingly.

5. The authors mention that a detection threshold must be set manually (P6 L48). Would this need to be done repeatedly as ambient light levels change throughout the night?

Thank you very much for picking up this particular point. The threshold is applied after a difference image is formed, hence slowly varying ambient light levels will not affect the threshold. In addition, field experiments are conducted overnight for up to 8 hours with images obtained inside a hut with closed doors and windows, hence ambient lighting levels do not change significantly in any case. An additional paragraph on this point was added on page 8.

6. If possible, it would be useful to compare the RRS system to standard backlit systems across light levels. For example, can a backlit system be just as good as an RRS system when there is more light? Or do e.g. saturation effects preclude this?

Thank you also for picking this point, additional clarification is warranted. The referee is correct that saturation effects preclude raising the light levels with the backlit system – due to the high central intensity that would then saturate. In contrast the more uniform illumination achieved with the RRS setup gives more consistent performance across the field of view. This has been clarified on page 12.

7. It would be useful to say a bit more about whether the tracking algorithm loses the mosquito when it lands for a few seconds. I imagine this algorithm would indeed lose the mosquito, since it operates on frame-to-frame differences.

The tracking algorithm used here is the same as that reported in [5]. Indeed, as frame to frame differences are used, a track terminates at a resting mosquito and new track would start if the mosquito subsequently moves. However, the algorithm described in [5] includes multiple processing passes which enable the end of one track to be connected to the start of another providing certain criteria (spatial and temporal bounds) are met. This detail has not been added here as it is fully described in the previous paper [5].

8. Is it possible at all to track the mosquito in the darkest region of the scene (i.e. in front of the volunteer)?

Both backlit and RRS systems rely on the contrast between the bright background and dark foreground (the mosquitoes). So, if there is an object that completely blocks light coming from the background (e.g. human host, bed, etc.), the segmentation algorithm cannot detect mosquitoes flying in front or behind such objects. A paragraph explaining this point was added on page 5.

9. I understand the practicality of having simple tracking algorithms for deployment in the field, but it would be good nonetheless to provide more comparison of this algorithm to state-of-the-art algorithms used in other insect flight-tracking studies. For example, see Straw, Andrew D., et al. "Multi-camera real-time three-dimensional tracking of multiple flying animals." *Journal of The Royal Society Interface* 8.56 (2010): 395-409.

The current submission's main focus is the illumination and segmentation aspects, however, the authors agree that further comparison with algorithms such as that in [22] are worthwhile and hence some additional content is provided in the introduction on page 3. As a general comment, unless some actively controlled intervention is connected to the output of the tracking system, there is no functional benefit in the algorithms operating in real time (with the possible exception that stored data files would be considerably smaller).

It is worth capturing that [22] describes a different class of tracking system (with different purpose) for lab based experiments, which are quite far from natural conditions. Moreover, [22] emphasises the identification of orientation of the flying animal which for mosquitoes would necessitate significantly higher magnification and hence reduced fields of view (defeating the objective of the entomology studies addressed here). As discussed in the manuscript, the authors are not aware of a successful tracking system which allows 3D tracking of a similarly sized scene and operating on small animals such as mosquitoes, especially, which can be used in remote field test sites (an initial report of such an approach is given in [26] and a more complete publication is expected in due course).

Comment from referee 2

This is a revised version of the original paper. I have the following concerns :

- Major point : I do not see clearly the difference between the RRS system and the backlit one. Authors must provide a figure (like figure 1) which shows clearly the two setup.

Whilst the aim of this submission is to introduce the RRS setup, we appreciate that a large part of the manuscript concerns a comparison between the two systems, hence it is appropriate to include a schematic of each approach side by side in Figure 1b.

Minor points: - model of the fresnel lens also supplier must be indicated

Fresnel lenses used are PMMA 1400x1050 mm fresnel lens with pitch of 0.112 mm and focal distance of 1200 mm made by NTKJ CO., LTD. (Japan) model CF1200 [24]. Information about the manufacturer and model added to the manuscript on page 4.

- were telecentric objectives used for the camera ? It is not clear. If so, please indicate the model and the optical parameter of the objectives used in this study.

There seems to be a bit of confusion here. The tracking system uses conventional camera lenses with fairly short focal distance (14 mm) in combination with the large aperture Fresnel lenses. Taken together, the camera + Fresnel lens form a telecentric imaging setup as the spacing between the lenses is equal to the sum of the focal lengths. The specific make and model of the camera lens depends on the specific camera used. At the moment, Nikon 14mm f2.8 D AF ED and Canon EF

14mm f2.8 L II USM lenses are utilised. Details on the camera lens make and model and a short clarification were added on page 5.

- how big was the surface covered by the tape of 3M material ? Please indicate the cost ?

The retroreflective material in the form of self adhesive tape is applied to a plywood board (size 2.4 m x 1.2 m). One board, oriented vertically, is used per camera. The cost depends on the supplier. Moreover, the manuscript is not making any particular claim about the cost of the system. So, adding details of the cost of the RRS material will look out of place in the manuscript. For the referee's benefit, the cost is about £600 per board. Details of the size and how the tape is used were added on page 4.

- authors must provide the code used for the image processing. The code must be made available on a GitHub for example.

The authors agree with the referee and the code for segmentation and tracking is being made available via GitHub [29]. Please, see the added note on page 16

- the attached video must be provided with explanations (comments). Subtitles have been added to the video to explain the important features revealed.

Comment from referee 3

The authors provide details of an improved technique to track free-flying nocturnal mosquitoes around bed nets in 2D. Although a highly specific, context dependent, research set-up, the technique is useful to answer fundamental questions about mosquito behavior around bed nets or can be used for some other vector arthropods. The manuscript is well written.

The manuscript is an 'improved-methods' paper, mainly by adding a retro-reflective screen to improve illumination conditions. Figure 9 in particular is an exemplar image of how the RRS system can be implemented, however it does not provide the analysed data.

The main focus of this submission is to describe how an RRS can improve the illumination conditions and together with improved algorithms for segmentation, increased reliability in extracting mosquito tracking can be obtained for the challenging case of a human baited bednet. Fully analysed data are included for this case from both the backlit and RRS imaging setups, thereby establishing quantitatively the performance benefit. Figure 9 was included as an initial tracking result from a proof of principle experiment to assess individual attractiveness to mosquitoes due to a referee comment from our previous submission to the Interface journal, where the referee felt that the bednet application was somewhat niche. The authors maintain that inclusion of Figure 9 demonstrates the flexibility and ease of use of the RRS setup and hence is relevant in this manuscript that is aimed at the improvements made to the tracking system. However, we have elected to report any significant biological findings separately, e.g. reference [4] from the human baited bednet study that were supported by the RRS imaging approach. A full study of individual attractiveness to mosquitoes is beyond the scope of the current submission, however, the reported image demonstrates that RRS imaging is a suitable tool to quantify such human-mosquito interactions.

The authors focus on the relevant 2D data they obtained earlier, using a previous version of the system, but seem to ignore other research/techniques in the lab or semi-field where mosquitoes were successfully tracked under challenging conditions and where flight paths were reconstructed in 3D.

As it was emphasised both in the manuscript itself and the response to the referees appointed by the Interface journal, the authors are not aware of any other tracking systems capable of either 2D or 3D tracking at the same scale and for recordings inside sub-Saharan Africa dwellings and which can handle similar temporal scale (up to 8 hours). The authors have included what we feel are the most appropriate references to previous reports of mosquito tracking systems (and other behavioural studies). If the referee is aware of additional references for systems having similar capabilities then we would be grateful for those references.

The introduction (p4) and proposed methods focuses on back-lit tracking and offline acquisition. There are good reasons for choosing this method, but the authors could highlight,

or discuss, that other illumination methods are possible too and that online tracking would tackle their challenge with processing large data files and data handling.

The authors agree with the referee's comment. Additional information has been added around alternative illumination approaches (page 2 in the introduction). There is inevitably a compromise in achieving contrast across the whole field of view and irrespective of the mosquito's position within the measurement volume. With regard to online processing, then yes in principle if this could be achieved robustly it would save storage of large data volumes. Part of the author's rationale in saving all data for later analysis is that there can be instances where the human bait or other external artefact causes significant changes and hence there can be many thousands of objects that suddenly appear (bednet movement or the suspending strings) some of which can be similarly sized to the mosquitoes. Maintaining the integrity of the temporal image sequence is essential for resolving conflicts in tracking and hence we have always favoured storage of the full image stream for later analysis. Some additional detail has been added.

I concur with many of the original comments made by reviewer #1. I understand from the authors' reply that the improved method (adding RRS) was already implemented during the studies of the recently published ref.4. This would indeed be an argument to have the updated, detailed methods, included within that paper, rather than publishing a separate methods paper.

Whilst we appreciate the referee's opinion on including the RRS method detail within the recent [4], this is no longer possible and had been accepted for publication before the submission of this manuscript. Further, it would then have been more difficult to show the potential use of the RRS approach in dealing with other entomology questions.

Additional minor comments/suggestions: - p3, line 25: most? There are many more mosquito-behavioural activities to quantify I can think of

Unfortunately, the sentence

"In order to understand better how hungry host-seeking mosquitoes are affected by the bednet, or how net alterations (physical as well as chemical) might improve net performance, entomologists are most interested in visualizing activity around the sleeping space, the bednet suspended above it and the regions around this in order to examine details of approach, attack and departure [11, 12]."

caused some confusion. It did not mean to state that approach, attack and departure around a bednet are the only mosquito behaviours that entomologists are interested in. We appreciate that mosquitoes have a whole lifecycle and vector biologists are interested in behaviours of all aspects of their lives. What this sentence meant to convey was that entomologists are mostly interested in approach, attack and departure behaviour around bednets as this is a point in their lifecycle where their position is known and is also directly on the pathway for disease transmission. To avoid confusion, the decision was made to split this long sentence into two to make it easier to read as follows:

"Entomologists need to better understand how hungry host-seeking mosquitoes are affected by the bednet, or how net alterations (physical as well as chemical) might improve net performance. For that, they are particularly interested in visualising activity around the sleeping space, the bednet suspended above it and the regions around this in order to examine details of approach, attack and departure [11, 12]."

See changed text in page 2.

- Reference #4 is published now right? (remove accepted, complete reference)

Yes, it was published after this revised manuscript was submitted. The reference was corrected to the full reference to the published version, see [4]

- p5, line 23: tracking ..

Thank you for noting. Corrected this and several other double points.

- p10, line 38: spelling coluzzii (no capital, double i)

Thanks for noting. Corrected, see at page 11.